# OXPHOS remodeling in high-grade prostate cancer involves mtDNA mutations and increased succinate oxidation

Bernd Schöpf [1], Hansi Weissensteiner [1], Georg Schäfer[2], Federica Fazzini[1], Pornpimol Charoentong[3], Andreas Naschberger[1], Bernhard Rupp[1], Liane Fendt[1], Valesca Bukur[4], Irina Giese[4], Patrick Sorn[4], Ana Carolina Sant'Anna-Silva [5], Javier Iglesias-Gonzalez[6], Ugur Sahin[4], Florian Kronenberg [1], Erich Gnaiger [5,6] & Helmut Klocker [7✉]

Rewiring of energy metabolism and adaptation of mitochondria are considered to impact on prostate cancer development and progression. Here, we report on mitochondrial respiration, DNA mutations and gene expression in paired benign/malignant human prostate tissue samples. Results reveal reduced respiratory capacities with NADH-pathway substrates glutamate and malate in malignant tissue and a significant metabolic shift towards higher succinate oxidation, particularly in high-grade tumors. The load of potentially deleterious mitochondrial-DNA mutations is higher in tumors and associated with unfavorable risk factors. High levels of potentially deleterious mutations in mitochondrial Complex I-encoding genes are associated with a 70% reduction in NADH-pathway capacity and compensation by increased succinate-pathway capacity. Structural analyses of these mutations reveal amino acid alterations leading to potentially deleterious effects on Complex I, supporting a causal relationship. A metagene signature extracted from the transcriptome of tumor samples exhibiting a severe mitochondrial phenotype enables identification of tumors with shorter survival times.

[1] Institute of Genetic Epidemiology, Department of Genetics and Pharmacology, Medical University Innsbruck, Schöpfstraße 41, A-6020 Innsbruck, Austria. [2] Institute of Pathology, Neuropathology and Molecular Pathology, Medical University Innsbruck, Müllerstraße 44, A-6020 Innsbruck, Austria. [3] Department of Medical Oncology, National Center for Tumor Diseases, University Hospital and German Cancer Research Center (DKFZ) Heidelberg, Im Neuenheimer Feld 267, D-69120 Heidelberg, Germany. [4] TRON, Translationale Onkologie an der Universitätsmedizin der Johannes-Gutenberg-Universität Mainz gGmbH, Freiligrathstraße 12, D-55131 Mainz, Germany. [5] Department of Visceral, Transplant and Thoracic Surgery, D. Swarovski Research Laboratory, Medical University Innsbruck, Innrain 66/6, A-6020 Innsbruck, Austria. [6] Oroboros Instruments GmbH, Schöpfstraße 18, A-6020 Innsbruck, Austria. [7] University Hospital for Urology, Division of Experimental Urology, Department of Surgery, Medical University Innsbruck, Anichstraße 35, A-6020 Innsbruck, Austria. ✉email: helmut.klocker@i-med.ac.at

Prostate cancer (PCa) is the most prevalent male non-cutaneous cancer type in Western countries, accounting for an estimated 10% of all cancer related deaths in Europe[1]. Recent studies suggest a multifactorial etiology encompassing an accumulation of genetic and epigenetic aberrations[2]. Although primary PCa has been extensively studied only few genomic aberrations including PTEN deletion, TMPRSS2-ERG fusions, and CDKN1B deletion but no driver mutations have been found[3]. Among other carcinogenic alterations, adaptations in metabolism and energy turnover might contribute to PCa formation and progression.

Alterations in mitochondrial (mt) metabolism including oxidative phosphorylation (OXPHOS) are a hallmark of cancer[4]. Mitochondria play an important role during tumorigenesis by orchestrating cellular energy transformation, apoptosis, and reactive oxygen species (ROS) signaling[5]. The bulk of cellular ATP is produced in the mitochondria by the stepwise oxidation of substrates via the tricarboxylic acid (TCA) cycle. Electrons are fueled into the mt-electron transfer system (ETS), catalyzed by NADH and succinate-linked dehydrogenases in the mt-matrix and mt-inner membrane. Translocation of protons generates an electrochemical potential difference across the mt-inner membrane, which is used by ATP synthase.

While the majority of ETS machinery proteins are encoded by nuclear DNA (nDNA), 13 ETS subunits are encoded by mtDNA, a small circular genome. mtDNA mutations have been linked to PCa formation and progression[6,7]. Compared to nDNA, mtDNA exhibits a higher mutation rate caused by increased exposure to ETS-derived ROS and less efficient DNA repair[8]. Random segregation and subpopulation replication of mtDNA variants lead to "heteroplasmy" (HP), the presence of different populations of mtDNA variants in a mitochondrion, cell or tissue. mtDNA mutations are frequently found in localized PCa[9–12]; however, their functional consequences remain elusive.

Only limited data on TCA cycle and OXPHOS activity in primary PCa tissue is available and little is known about specific malignant PCa metabolism[13]. Understanding the impact of mtDNA mutations might help to characterize metabolic adaptations exploited to drive PCa formation and progression. In this study, we aim to unravel the interplay between mtDNA mutations, the expression and function of mitochondrial enzymes and the metabolic PCa phenotype to identify specific mitochondrial cancer signatures guiding towards new approaches for therapeutic intervention. Our results reveal a shift towards higher oxidation of succinate, which is associated with deleterious mutations in mitochondrial Complex I genes and a rewired expression of mitochondrial metabolic enzymes in primary prostate cancer.

## Results

**Shift to succinate-driven OXPHOS in prostate tumors.** Paired benign and malignant prostate tissue samples were isolated after radical prostatectomy from fifty PCa patients. Benign samples were taken distant from the tumor to minimize field effects (Fig. 1a). Each biopsy was split for histopathological diagnostics (Fig. 1b) and high-resolution respirometry (HRR). See Table 1 for patient and tumor characteristics. OXPHOS was analyzed simultaneously in paired benign/malignant tissue biopsies of each prostate specimen by sequentially assessing respiratory coupling control and capacities of single and combined mt-electron transfer (ET) pathways[14] (Fig. 2a–c, Supplementary Tables 1–2). A short-term treatment with $H_2O_2$ was included to simulate oxidative stress.

Benign tissues showed a significantly higher NADH-pathway OXPHOS capacity (N; electron entry to the Q junction via CI) supported by glutamate&malate, $GM_P$, and pyruvate&glutamate&malate, N ($PGM_P$), respectively, compared to PCa tissue ($p = 2 \times 10^{-5}$ and $p = 8 \times 10^{-6}$, two-sided paired-samples $t$-test, Fig. 2d). However, upon further addition of succinate (S), no differences in the OXPHOS ($NS_P$) and ET ($NS_E$) respiratory capacities were observed, indicating compensation of the N-pathway deficiency by increased convergent electron transfer through the succinate-pathway (electron entry to the Q junction via CII). Full restoration of $NS_P$ and $NS_E$ capacities in tumor tissue was driven largely by succinate and to a smaller extent by pyruvate. Glutamate&malate-driven OXPHOS by addition of ADP triggered an increase of $O_2$ flux of 2.4 pmol s$^{-1}$ mg$^{-1}$ in tumor compared to 4.5 pmol s$^{-1}$ mg$^{-1}$ in benign tissue samples (Fig. 2e). Addition of pyruvate and succinate, respectively, elicited

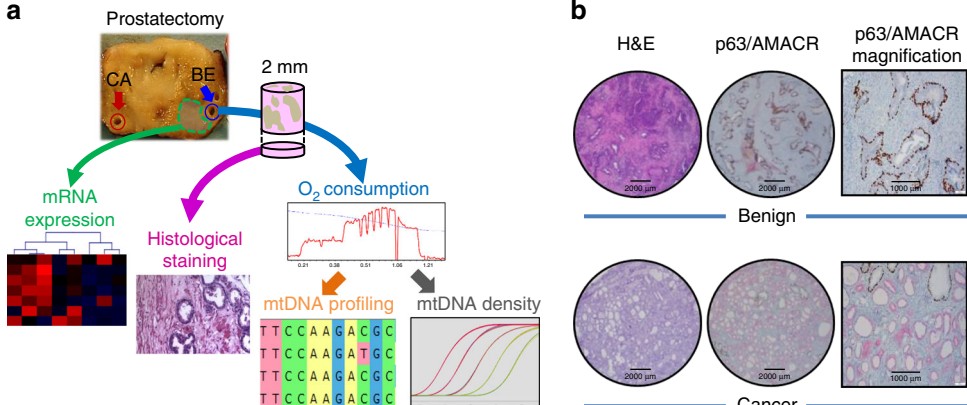

**Fig. 1 Sample workflow and tissue sample confirmation. a** From each of 50 radical prostatectomy specimens a tumor and a non-malignant benign tissue punch needle biopsy was extracted by an experienced uropathologist from contralateral sites of the specimens (blue/red circles). While a small portion of the extracted tissue cores was fixed and used for histological stains (pink arrow) and confirmation of tissue identity, the rest was used immediately for high-resolution respirometry (HRR, blue arrow), subsequent NGS mtDNA profiling (orange arrow) and mtDNA copy number determination (gray arrow). From 16 cases of this cohort, frozen tissue samples directly adjacent to the biopsy cores (green area) were isolated by macrodissection followed by RNA extraction for gene expression profiling via total RNA-NGS (green arrow). **b** Representative hematoxylin and eosin staining (H&E) and p63 (non-malignant tissue marker, brown)/AMACR (malignant cell marker, red) double-immunostaining (p63/AMACR) of fixed and paraffin-embedded benign and malignant prostate tissue samples extracted for HRR. One representative of 50 cases is shown. Scale bars indicate 2000 μm (H&E, P63/AMACRA stains) and 1000 μm (P63/AMACR higher magnification), respectively.

**Table 1 Patient and tissue sample characteristics.**

| Patient and tissue sample characteristics | |
| --- | --- |
| Age [a] | 62.7 ± 8.0 (39.9–73.4) |
| PSA (ng mL$^{-1}$) | 12.7 ± 27.2 (2.0–181.6) |
| fPSA (%) | 14.3 ± 5.8 (3.6–28.1) |
| Prostate weight (g) | 44 ± 14 (20–79) |
| *Pathological stage, N (%)* | |
| pT2 (localized) | 25 (50.0) |
| pT3 (extracapsular extension) | 21 (42.0) |
| pT4 (invasion into adjacent structures) | 4 (8.0) |
| *Tumor histology PCa tissue samples, N (%)* | |
| Gleason score 6 (patterns 3 + 3) | 11 (22.0) |
| Gleason score 7 (patterns 3 + 4) | 9 (18.0) |
| Gleason score 7 (patterns 4 + 3) | 21 (42.0) |
| Gleason score 8 (patterns 5 + 3) | 1 (2.0) |
| Gleason score 9 (patterns 4 + 5) | 6 (12.0) |
| Gleason score 10 (patterns 5 + 5) | 2 (4.0) |
| *Tissue sample wet mass (mg)* | |
| Benign | 6.49 ± 1.48 |
| Malignant | 6.51 ± 1.46 |

Paired malignant and non-malignant tissue biopsy samples were extracted from the radical prostatectomy specimens by an experienced uropathologist. Age, total serum prostate specific antigen (PSA) and percentage of free PSA (fPSA) at the time of tumor diagnosis. Tissue biopsy wet mass used in high-resolution respirometry (HRR) was determined prior to the transfer of permeabilized tissue samples into the O2k chambers. Tumor histology characteristics of radical prostatectomy specimens (Gleason scores of tissue samples and pathological stages) were determined using routine histopathological procedures. Data represent mean ± SD, median (age), range or quantity and percentage, respectively.

significantly higher increases of $O_2$ flux in malignant compared to benign samples ($N_P$ minus $GM_P$, $NS_P$ minus $N_P$) and thus recovered full respiratory capacity in tumors (Fig. 2e). Finally, CI inhibition by rotenone confirmed significantly higher N-pathway capacity of benign compared to tumor tissue ($NS_E$ minus $S_E$, Fig. 2e). Oxidative stress resulted in a reduction of $O_2$ flux in both tissue types, however, significantly more in the tumor tissue (1.7 vs. 1.1 pmol s$^{-1}$ mg$^{-1}$, $p = 0.004$, two-sided paired-samples $t$-test, Fig. 2e, $GM_P$ minus $GM_{P,pre}$).

Differences in OXPHOS capacities were mainly driven by high-grade tumors (Gleason score >7), Fig. 2f, g). Whereas relative respiration (FCR) of low-grade tumors (Gleason score ≤7) was similar to benign tissue, it was significantly different in the high-grade tumors (Fig. 2f). A direct comparison of specific $O_2$ fluxes confirmed a significant reduction of N-pathway respiration (Fig. 2g—effects of ADP and rotenone) and increase of pyruvate and especially succinate respiration (Fig. 2g—effects of pyruvate and succinate) in the high-grade tumors. Altogether, HRR analysis of paired benign/cancer tissue samples uncovered a substrate shift from glutamate&malate to succinate and to a lesser extent pyruvate in tumors, particularly in the high-grade tumors, and higher vulnerability of tumors by oxidative stress.

Several control experiments were performed. Split tissue samples ($N = 6$) analyzed with or without the oxidative stress step indicated a consistent difference through all post-treatment states (Supplementary Fig. 1a). $O_2$ flux differences were reduced only for the oxidative stress step (Supplementary Fig. 1b) and analysis of N- and NS-pathway capacities without prior $H_2O_2$ treatment confirmed the differences between benign and tumor samples (Supplementary Fig. 1c–d). Although the extent of inhibition of $O_2$ flux by oxidative stress was higher in tumor tissue, it remained consistent through all substrate/coupling states within the benign and the malignant samples compared to untreated samples (Supplementary Fig. 1e). Finally, measurement of the single enzymatic step of CIV indicated no selective inhibition by $H_2O_2$ (Supplementary Fig. 1f, compare to Supplementary Fig. 1e—BE). Comparison of benign/benign

paired samples confirmed that the observed respiratory differences between benign and malignant samples were not caused by tissue heterogeneity or sampling bias. Twenty paired benign/benign samples subjected to the same analysis protocol as benign/malignant samples showed no differences (Supplementary Fig. 2a–c). There were also no differences between zonal origins of benign samples (medial or posterior peripheral zone or transitional zone) (Supplementary Fig. 2c–e).

**Increased mtDNA mutation load in primary human PCa tissue.** We sequenced the mtDNA genome of all tissue samples recovered from HRR analysis and assessed a potential link of respiratory patterns and HPs in ET-protein or tRNA genes. Average mtDNA sequence coverage was >10.000 fold (Supplementary Fig. 3). Mutations with HP levels of higher 2% were considered for further analysis. Details of all 147 HPs found across the sample pool, including their presence and pathological context listed in the MitImpact 2.9 database[15], are presented in Supplementary Data 1. Private mutations, defined as HPs found only in one tissue type, were more frequent in the malignant samples (Fig. 3a). In benign samples 33 private HPs were found compared to 84 in the malignant tissues. Significantly more HPs were located in the mtDNA coding region of cancer tissue (72 vs 23, odds ration=2.7, $p = 0.007$, Fisher's exact test), whereas HP frequency in the non-coding control region was similar in benign and cancer tissues (10 vs 12) reflecting an accumulation of HPs in the coding mt-genome of the tumors (Fig. 3b). Additionally a higher frequency of non-synonymous HPs in all ET-protein-coding genes (*MT-ND1-6* = CI, *MT-CO1-3* = CIII, *MT-CYB* = CIII and *MT-ATP6-8* = $F_0F_1$-ATPase) was detected in the malignant tissue samples (38 vs 10) while the number of mt-tRNA mutations (*MT-T* = tRNA) was the same (2 vs 2, Fig. 3c). Thirty five HP sites in the malignant samples showed an allele frequency (AF) > 10% compared to only 11 in the benign samples, while 15 of the HPs detected in the malignant samples showed an AF > 50% compared to only 5 in the benign samples (Fig. 3d). Consequently, overall HP levels were significantly higher in malignant compared to benign tissue samples ($p = 0.03$, Wilcoxon rank-sum test, Fig. 3e). Almost half (46%) of the tumor samples harbored ≥2 HPs while 60% of the benign samples harbored no HP (Fig. 3f). The number of HPs per mt-gene was correlated to gene size as shown for MT-ND genes (Pearson's $r = 0.93$, $p = 0.004$, two-tailed $t$-test, Fig. 3g), suggesting no cancer-specific mutational "hotspots" in primary PCa.

Of all non-synonymous mutations in protein coding genes, only two variants (*T15719C* and *T10551C*) have been previously detected in human cancers and only two variants were associated with a specific clinical phenotype according to MitoMap, maternally inherited diabetes-deafness syndrome (MIDD; *G3421A*) and developmental delay, seizure and hypotonia (*G4142A*) (Supplementary Data 1). Only the *T10551C* mutation (S28P in ND4L protein), exhibited a high HP level (58%) while the allel frequencies of all other variants was below 20% in our samples. Of the four mutations in mt-tRNA (*MT-T*) genes (Supplementary Data 1, Supplementary Fig. 4a), one variant (*G15995A*, *MT-TP*, tRNA$^{Pro}$) has been previously detected in a cystic fibrosis patient[16] and was classified as "likely pathogenic" by MitoTIP ([https://www.mitomap.org/cgi-bin/mitotip]). This mutation leads to a $G > A$ substitution in a highly conserved region of the anticodon-stem, resulting in a base-pair mismatch with likely effects on RNA folding and stability (Supplementary Fig. 4b). However, the frequency of this tRNA variant (15%) most certainly does not impact mt-function[17].

To evaluate the functional relevance of the mtDNA variants we determined the MutPred Pathogenicity score[18,19] for all

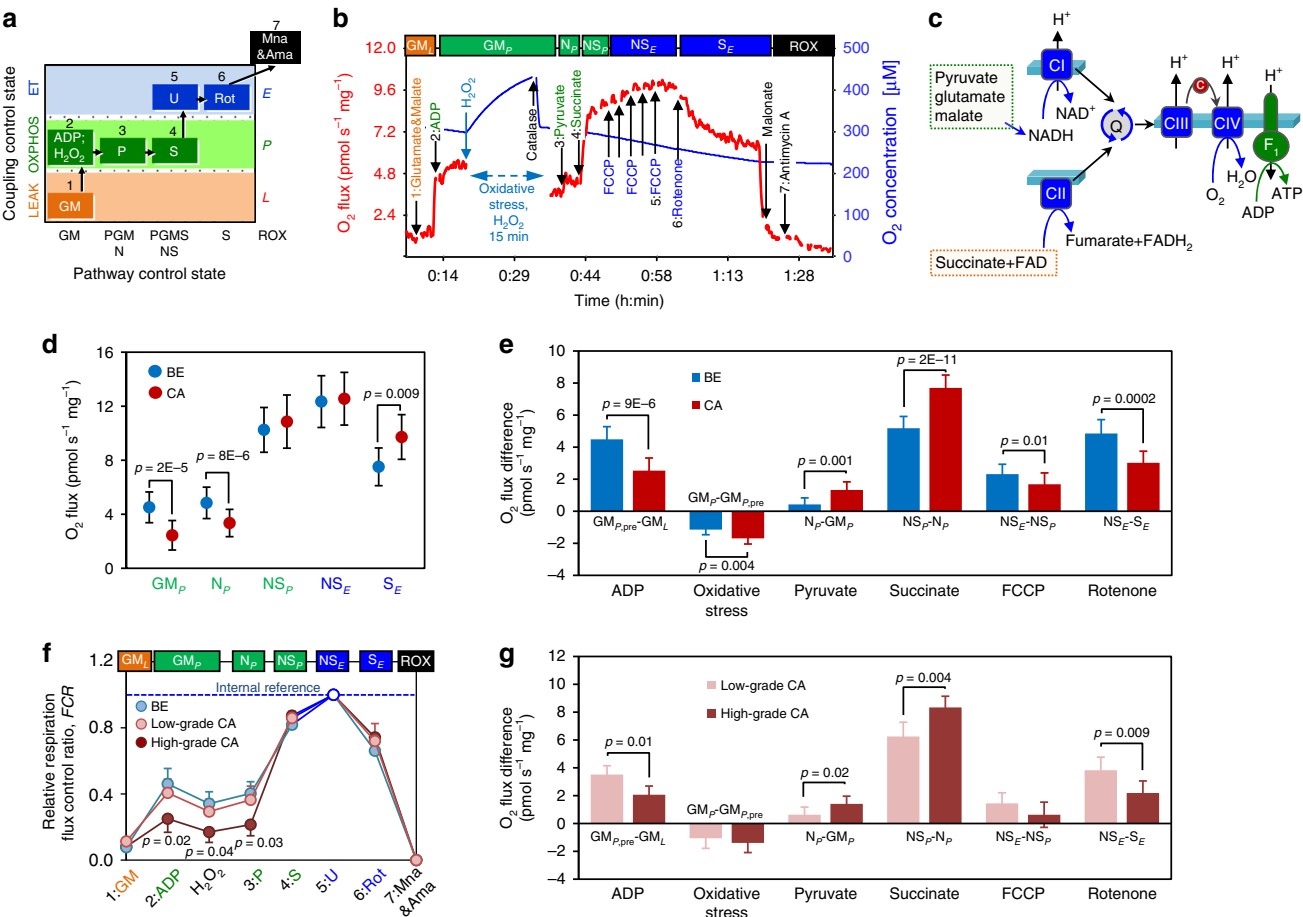

**Fig. 2 High-resolution respirometry of prostate tissue samples. a** Coupling/pathway control diagram showing the sequential steps in the substrate-uncoupler-inhibitor titration (SUIT) protocol with different coupling states. See Supplementary Tables 1–2 for HRR protocol details. **b** Representative HRR traces with permeabilized tissue. Red line (left $Y$-axis): wet mass-specific $O_2$ flux (oxygen consumption (pmol s$^1$ mg$^{-1}$)). Blue line (right $Y$-axis): $O_2$ concentration [μM]. Substrate-uncoupler-inhibitor titrations are indicated by arrows. Different coupling/pathway control states are indicated in boxes: LEAK (orange); OXPHOS (green); ET (blue); ROX (black). **c** Schematic representation of mitochondrial electron transfer from NADH-linked substrates through Complexes CI, CIII, and CIV (N-pathway) and from succinate through CII, CIII, and CIV (S-pathway). G, M and P support the N-pathway through CI into the Q-cycle (Q); succinate provides electrons via CII into the Q junction. Electrons are transferred from CIII via cytochrome $c$ (c) to CIV where $O_2$ is reduced to $H_2O$. $H^+$ ions are pumped across the mt-inner membrane by CI, CIII, and CIV to generate an electrochemical potential difference across the mt-inner membrane, which drives phosphorylation of ADP to ATP by $F_OF_1$-ATPase. **d** Respiratory capacity in benign (blue, $N = 50$) versus malignant (red, $N = 50$) tissue samples: OXPHOS-capacity ($GM_P$, $N_P$ and $NS_P$) and ET-capacity ($NS_E$ and $S_E$). **e** Effects of substrates GM, pyruvate, succinate, oxidative stress, uncoupler FCCP, and CI inhibitor rotenone on $O_2$ flux in benign (blue, $N = 50$) and malignant (red, $N = 50$) tissue samples. **f**. Normalized respiratory capacities of high-grade tumor (Gleason > 7; dark red; $N = 10$) and low-grade tumor (Gleason ≤ 7, light red, $N = 40$) compared to benign samples (blue, $N = 50$). **g** Effects of substrates, oxidative stress, uncoupler, and CI inhibitor on $O_2$ flux in low-grade (light red, $N = 40$) and high-grade (dark red, $N = 10$) tissue samples. Data in (d–g) are presented as mean values ± SD. Statistical differences were tested using two-tailed paired-samples test (**d–e**), one-way ANOVA followed by Tukey's HSD (**f**) or Wilcoxon rank-sum test (**g**), respectively. Correction for multiple testing was performed using the Bonferroni-Holm procedure. Source data are provided as a Source Data file.

non-synonymous HPs (Fig. 3h). While only 14% of HPs of benign samples exhibited a high MutPred score (>0.75), half of the HPs of malignant samples fell into this category (Fig. 3i).

**Non-synonymous mtDNA mutations in high-grade tumors**. To evaluate a correlation of clinical parameters and mtDNA mutation frequency a logistic regression analysis was performed (Supplementary Table 3). Overall mtDNA mutation load correlated significantly with increasing patient age ($p = 0.04$, likelihood-ratio test), in line with recent reports[9,20,21] and with lower free to total PSA ratios (fPSA%, $p = 0.05$, likelihood-ratio test, Supplementary Table 3). A low fPSA% value is a prognostic indicator of poor prognosis[22]. While a correlation trend was detected for the pathological tumor (pT) stage ($p = 0.09$, likelihood-ratio test), we found no significant association with

histological tumor (Gleason) grade or total serum PSA, contrary to previous findings[23]. When looking only at non-synonymous mtDNA mutations, however, their frequency correlated with high-grade staging (Gleason > 7, $p = 0.02$, likelihood-ratio test) and again with lower fPSA% ($p = 0.05$, likelihood-ratio test). Thus, non-synonymous HPs with a potential impact on ET-function tended to accumulate in younger patients with unfavorable clinical characteristics.

**Deleterious mtDNA mutation load and OXPHOS capacity**. While increasing numbers of potentially deleterious mtDNA mutations in cancer samples are reported[9–12,21,23], their functional significance is rarely evaluated. We correlated the frequency and type of mtDNA mutations with functional data for each sample. Non-synonymous mutations in genes relevant for

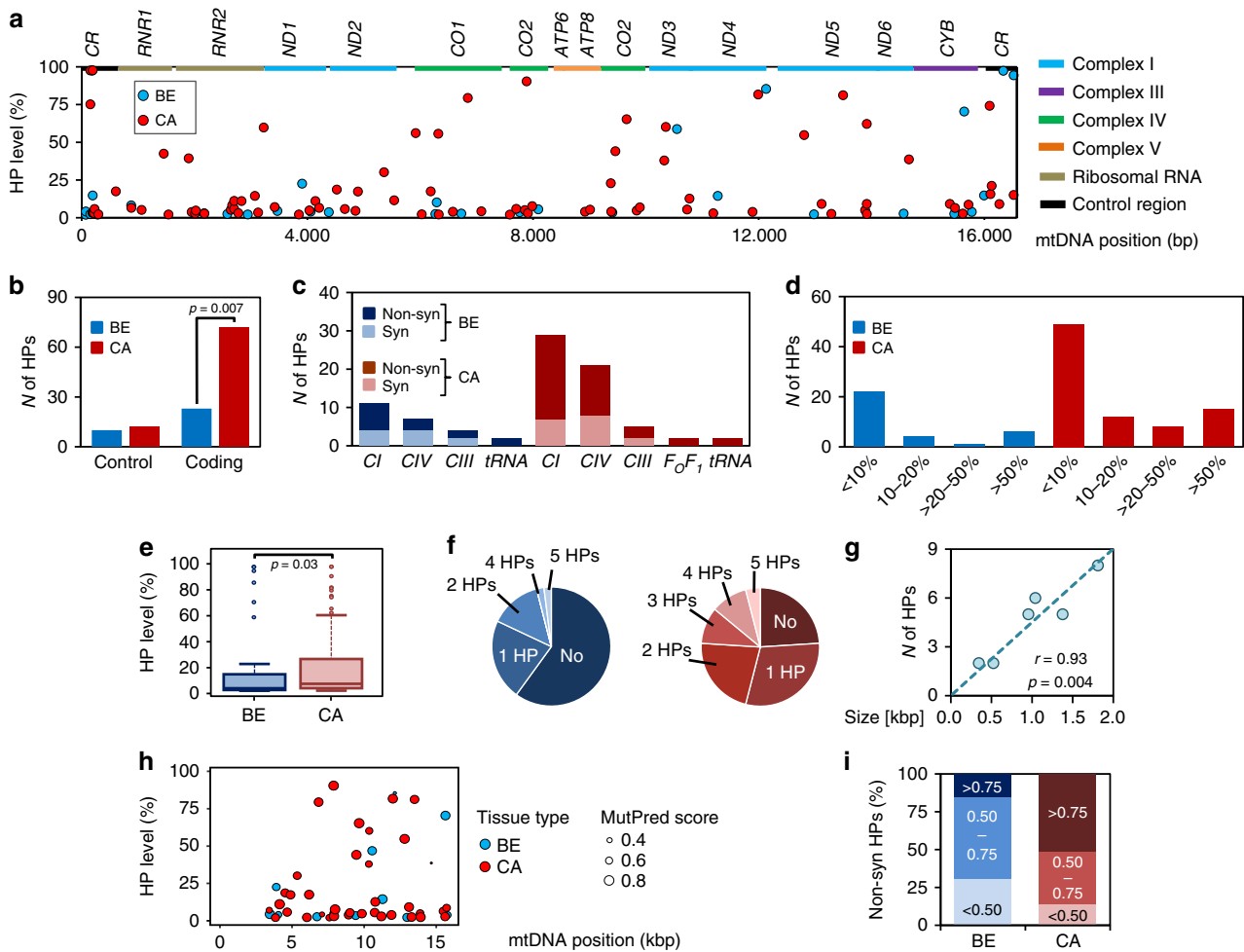

**Fig. 3 mtDNA heteroplasmies in benign and malignant tissue samples. a** Levels and locations of HPs in benign (blue, $N = 50$) and malignant (red, $N = 50$) samples across indicated loci of the mtDNA. The regions shown include the non-coding control region (CR) and the 13 protein coding genes, marked by colored boxes. ND, CO and ATP refer to genes coding for subunits of Complex I (CI; NADH:ubiquinone oxidoreductase), Complex IV (CIV; ferrocytochrome c:oxygen oxidoreductase and ATP synthase, respectively, whereas CYB encodes a subunit of Complex III (CIII; ubiquinol:ferricytochrome c oxidoreductase). **b** Total cumulative count of private mutations, located in either the non-coding D-loop or coding areas of the mt-genome in benign (blue, $N = 50$) and malignant (red, $N = 50$) tissue, respectively. Differences were tested for significance using Fisher's exact test. **c** Incidence of synonymous (Syn) vs non-synonymous (Non-syn) HPs located in protein coding sequences ($F_O F_1$-ATP synthase=CV; tRNA genes=tRNA) found in either benign (blue, $N = 50$) or malignant (red, $N = 50$) tissue samples. **d** Detailed proportions of samples harboring variants with defined HP levels (<10%, 10–20%, 20–50% and >50%) in the benign (blue, $N = 50$) and malignant (red, $N = 50$) samples. **e** Boxplot of HP levels comprising all heteroplasmies in benign and malignant tissue samples. Data are presented as boxplots indicating median, 25–75th percentile (box) and median ± 1.5IQR (whiskers), minimum and maximum values (dots). Differences in mean values were tested for significance using Wilcoxon rank-sum test. **f** Numbers of HPs found per sample in benign (blue circle, $N = 50$) and the malignant (red circle, $N = 50$) samples. **g** Correlation of gene size in kbp to HP count as detected in the mt-ND genes of malignant samples ($N = 50$). Linear correlation was established using Pearson's correlation coefficient (r) while correlation was tested for significance using two-tailed t-test. **h** MutPred Pathocenicity Scores of all non-synonymous HPs in benign (blue, $N = 50$) and malignant samples (red, $N = 50$) identified across the mtDNA. The size of the spots indicates the likely functional effects. **i** Proportion of benign (blue, $N = 50$) and malignant (red, $N = 50$) samples carrying HPs with defined MutPred Pathogenicity scores reflecting their likely functional effect. Source data are provided as a Source Data file.

the assembly or function of the OXPHOS machinery (*MT-ND*, *MT-CO* and *MT-CYB* genes) were considered as potentially deleterious. Samples carrying such mutations showed a significant decrease of relative GM-pathway capacity in both benign and malignant samples ($p = 0.005$ and 0.004, respectively, Wilcoxon rank-sum test, Fig. 4a). In contrast, mutations in the D-Loop exerted no effects (Fig. 4b). GM-pathway capacity was reduced only in samples carrying non-synonymous mutations in CI genes (*MT-ND1-6*) but not in those with CIII or $F_O F_1$-ATPase (*COI-III*, *CYB* and *ATP6-8*) gene mutations ($p = 0.0004$; one-way ANOVA followed by Tukey's HSD test, Fig. 4c).

Even moderate levels of deleterious mtDNA mutations in CI genes can lead to significant functional effects[24,25]. To evaluate

the impact of increasing HP levels on CI-function in detail, we divided malignant samples into subgroups exhibiting a HP level of either 30–60% or >60%. Higher HP levels were associated with a more pronounced reductions in respiratory N-pathway capacity (CI-dependent) and correspondingly with a higher proportion of S-pathway capacity to retain full aerobic ATP production (Fig. 4d). In the samples exhibiting the highest HP loads (>60%, $N = 4$), N-pathway capacity with glutamate&malate was reduced to 16% of total NS-OXPHOS capacity. Comparison of the respiratory capacities in samples harboring high levels of non-synonymous HPs in either CIV or CI genes only (Fig. 4f) showed similar N-pathway capacity while significantly higher S-pathway capacity was observed in the group harboring CI mutations

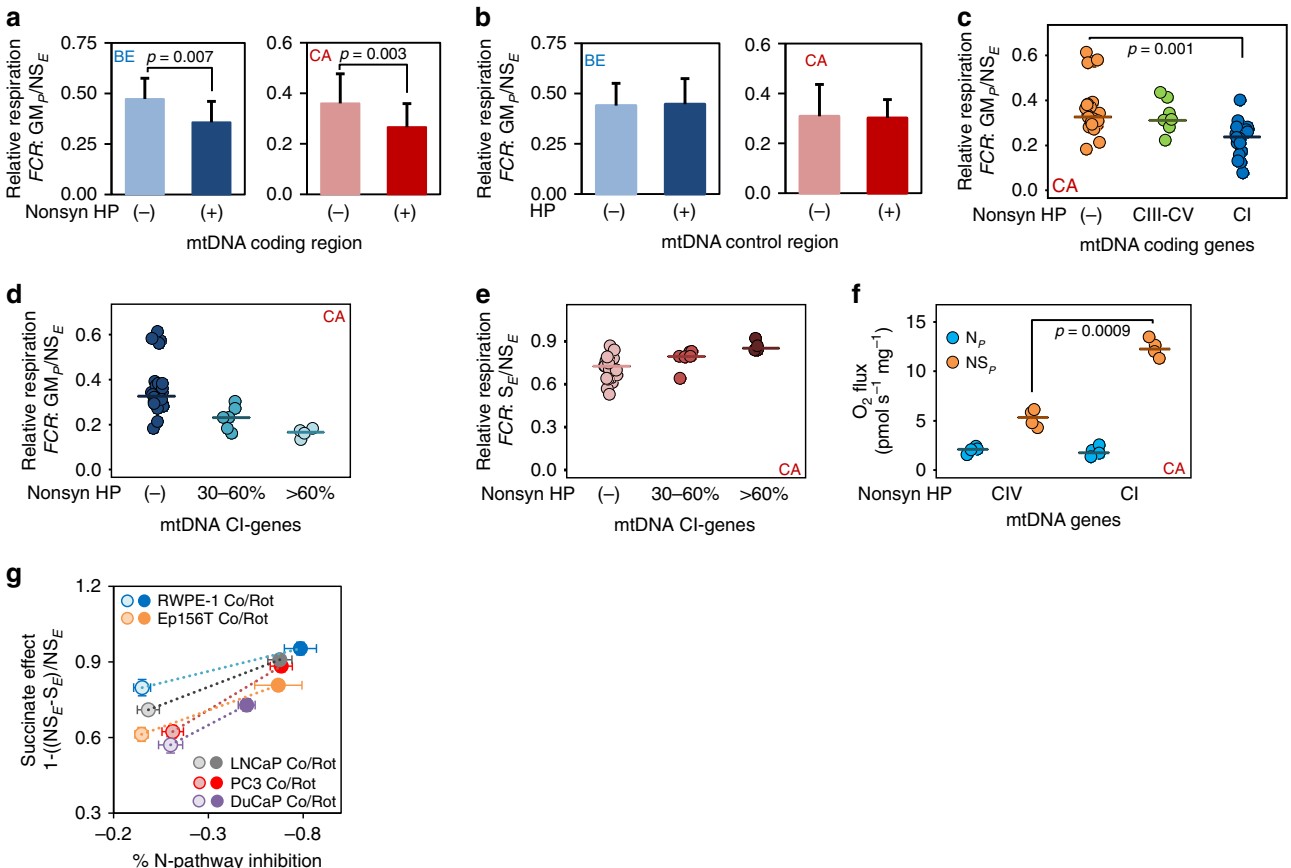

**Fig. 4 mtDNA heteroplasmies and respiratory capacities. a** Relative GM-OXPHOS respiratory capacity with glutamate&malate ($GM_P/NS_E$) in benign (blue) or malignant (red) samples carrying either no or only synonymous HPs (−) ($N_{BE} = 36$ and $N_{CA} = 23$) versus samples carrying non-synonymous HPs (+) ($N_{BE} = 14$ and $N_{CA} = 27$) in the coding regions of the mt-genome. Values present mean ± SD. Differences in mean values were tested for significance using Wilcoxon rank-sum test. **b** Comparison of relative GM-OXPHOS capacity in samples without mutations (−) ($N_{BE} = 38$ and $N_{CA} = 37$) versus samples with mutations (+) ($N_{BE} = 12$ and $N_{CA} = 13$) within the non-coding (control) region of the mt-genome. Values represent mean ± SD. **c** Impact of location of non-synonymous HPs on relative GM-OXPHOS capacities. Tumor tissue samples were categorized according to no HPs (−) (orange; $N = 24$), and HPs located in genes encoding proteins of CIII-CV (CO, ATP, CYB; green; $N = 10$) or in genes encoding proteins of CI (ND1–ND5; blue; $N = 20$). **d–e** Relative GM-OXPHOS ($GM_P/NS_E$, **d**) and relative S-ET ($S_E/NS_E$, **e**) respiratory capacities in malignant tissue samples harboring non-synonymous HPs in CI-coding mt-genes. Tumors were grouped into samples carrying no non-synonymous HPs ((−); $N = 23$), samples with variant levels of 30–60% (30–60%; $N = 6$) and samples with variant levels >60% (>60%; $N = 4$). Differences in mean values were tested for significance using one-way ANOVA followed by Tukey's HSD test. **f** N-pathway (blue) and S-pathway (orange) respiratory capacities in malignant samples harboring high-level (>60%) non-synonymous HPs in either CIV-coding genes ($N = 4$) or CI-coding genes ($N = 4$). Differences of in mean values were tested for significance using Wilcoxon rank-sum test. Values presented in c-f represent mean values and individual data points. **g** S-pathway OXPHOS capacity upregulation by partial inhibition of N-pathway oxidative flux in benign (RWPE1, $N = 3$; EP156T, $N = 3$) and malignant (PC3, $N = 6$; LNCaP, $N = 4$; DuCaP, $N = 3$, $N$ represents number of biologically independent experiments) prostate cell lines. Relative S-pathway OXPHOS capacity (normalized to total respiratory capacity, $NS_E$) with different degrees of N-pathway inhibition is shown for the five cell lines. Values represent mean ± SD. Source data are provided as a Source Data file.

($p = 0.002$, Wilcoxon rank-sum test) indicating that a defect in CIV will affect total respiration, while defects in CI can be sufficiently compensated.

To address whether reduction of N-pathway capacity triggers a respiratory pathway shift we subjected three malignant and two non-malignant prostate cell lines to HRR analysis and applied low concentrations of rotenone to partially inhibit NADH-linked respiration. In all cell lines partial inhibition ranging from 40 to 90% elicited a compensatory N→S-pathway shift in the OXPHOS state despite unchanged S-pathway capacity (Fig. 4g). Compensation ranged from 50 to more than 100% in the different cell lines. These results indicate mobilization of an S-pathway OXPHOS capacity reserve upon reduction of N-pathway oxidative flux in order to keep total OXPHOS capacity high.

**A structural basis for the effect of CI mutations**. To characterize the molecular and biochemical mechanisms by which the CI

mutations could affect respiration, an in-silico assessment of the potential structural changes in CI was performed, considering mutations exhibiting critical HP levels of ≥30% (Fig. 5a, b). We analyzed the structural impact of these mutations on the basis of the CI subunit in the recently published cryo-EM structure of the human respiratory supercomplex SC $I_2III_2IV_2$[26]. The mutations cluster near the charged central axis of CI (Fig. 5b, online supplement, ([http://genepi.i-med.ac.at/pca-mt-c1/]). That complex region is at the center of Q-cycle redox coupling and is crucial for translocation of protons across the mt-inner membrane[27].

To illustrate potential structural consequences of mutations, two examples exhibiting a very high (>80%) variant level and severe alteration of the HRR profile were examined in detail: the variant *T11991C*, recently also detected in a PCa tissue sample (COSM1132242, [https://cancer.sanger.ac.uk/cosmic/mutation/overview?id=1132242]) leads to substitution of a large hydrophobic, aromatic amino acid by a small, polar amino acid (F411S)

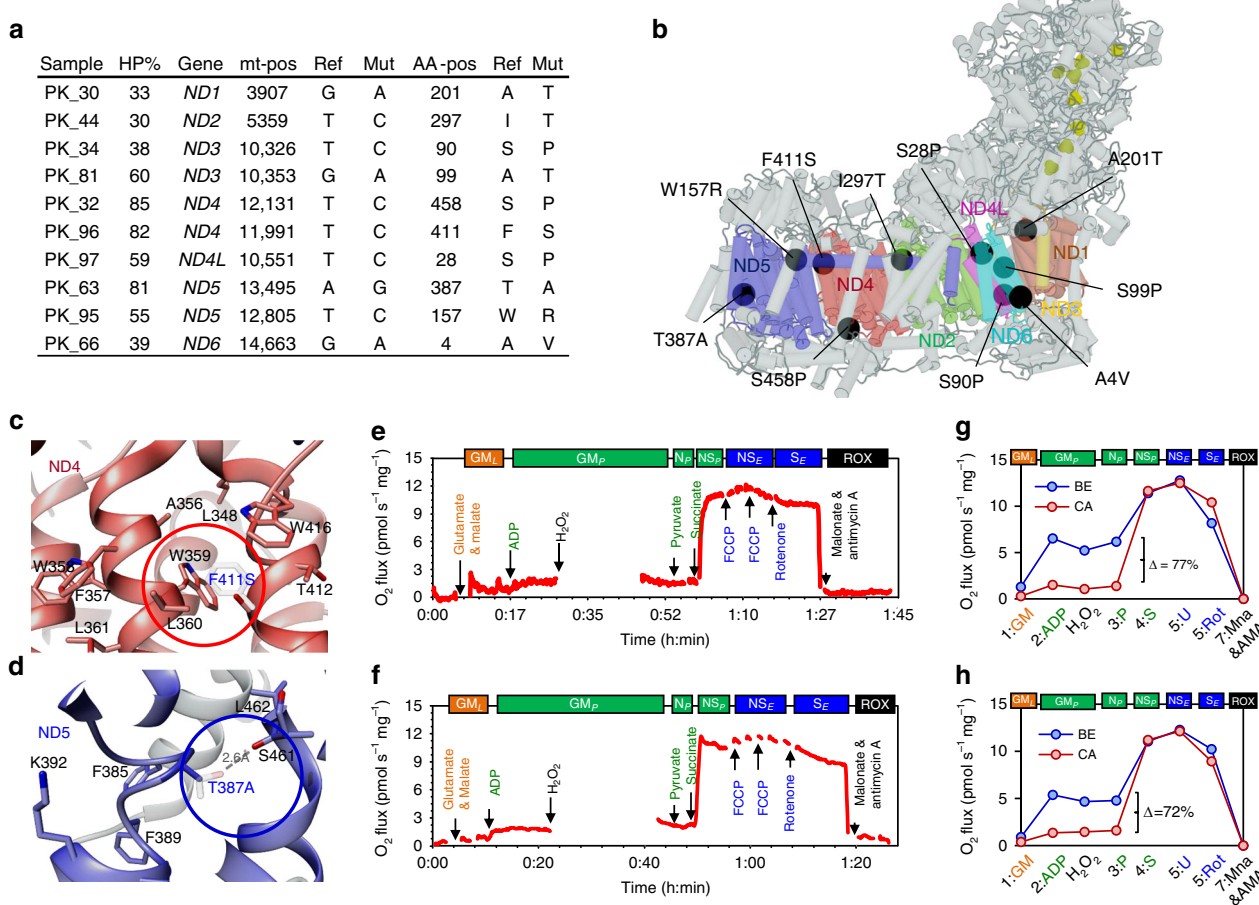

**Fig. 5 Structural impact of non-synonymous CI gene mutations. a** List of malignant samples carrying non-synonymous variants with HP levels ≥30% (HP % = heteroplasmy levels, Gene=mt-gene, mt-pos=location of variant on mtDNA sequence, Ref= nucleotide in rCRS sequence, Mut=altered nucleotide found in sample, AA-pos=position of encoded amino acid in protein sequence, Ref=amino-acid in wild-type protein, Mut=altered amino acid). **b** Structure of human respiratory Complex I and location of mutations found in this study. mtDNA-encoded CI-subunits (ND1-6) are shown in colors (ND1 = dark red, ND2 = light blue, ND3 = orange, ND4 = red, ND4L = purple, ND5 = dark blue and ND6 = light blue), mutations are shown as black circles and Fe-S clusters as yellow spheres. To visualize mutation sites see complemental iSee package using any JavaScript enabled web browser ([https://github.com/genepi/mt-c1]). **c** Structural change caused by the F411S mutation (red circle) in ND4. Wild-type (F) and mutated variant (S) amino acids are depicted and superimposed together with amino-acid residues involved in the hydrophobic interaction network of the α-helix structure (W359, L360, and W416, respectively). **d** Structural change caused by the T387A mutation located within the loop of the discontinuing helix 12 in the central axis of CI membrane domain (blue circle) in ND5. Wild-type (T) and mutated variant (A) are superimposed and the stabilizing hydrogen bond present in the wild-type protein is depicted by a scattered line. **e–f** HRR traces of the malignant biopsies carrying the F411S (e) or the T387A mutation (f), respectively. The red line represents wet mass-specific $O_2$ flux (oxygen consumption (pmol s$^{-1}$ mg$^{-1}$]). **g–h** Respiratory capacities of malignant samples (red) carrying either the F411S mutation (**g**) or the T387A mutation (**h**), compared to the corresponding benign tissue (blue). Values represent mean ± SD of the two separate measurements for each tissue sample. See Fig. 2 and Supplementary Tables 1–2 for abbreviations, coupling states and SUIT protocol.

interrupting a hydrophobic interaction network based on Pi-stacking between conserved helices in the ND4 domain of CI (Fig. 5c). The variant A13495G in the ND5 gene leads to the loss of a polar residue within the loop of the discontinued helix 12 in the central axis of the CI membrane domain (T387A). This part of the structure was annotated as flexible region that might play an important role initiating local conformational changes necessary to position corresponding key residues in the central axis (e.g. K392) during proton pumping[27]. A hydrogen bond is lost due to the T387A mutation in this important region (Fig. 5d). HRR analysis of these variant tumor samples revealed very low N-pathway capacity (PGM; pyruvate&glutamate&malate) <10% of combined NS-driven respiration compared to the corresponding benign sample (Fig. 5e, f). N-pathway capacities of the malignant biopsies harboring these variants were decreased by >70% (Fig. 5g, h), indicating a causal role.

**Increased mtDNA content and mt-mass in high-grade tumors**. Having found significant differences in mt-bioenergetics and mtDNA mutations between benign and malignant prostate tissue we examined whether mtDNA copy number (mt-CN) or mt-mass might be altered. Duplex mtDNA/nDNA PCR[28] revealed a median mt-CN load of 310 per nuclear genome (range: 172–698) in the benign and 303 (range: 143–645) in the malignant tissue samples, indicating no meaningful difference (Fig. 6a). While it has been proposed that mt-CN constitutes a reliable marker for mt-protein density and OXPHOS capacity in general[29], we found a negative correlation between mt-CN and OXPHOS capacity in benign and no correlation in malignant tissues ($r = -0.30$ and 0.057, $p = 0.03$ and 0.7; respectively, Fischer's exact test, Fig. 6b). PCa samples harboring potentially deleterious mtDNA mutations had significantly higher mt-CNs compared to samples carrying no HPs ($p = 0.03$; Wilcoxon rank-sum test, Fig. 6c). mt-CN ratios

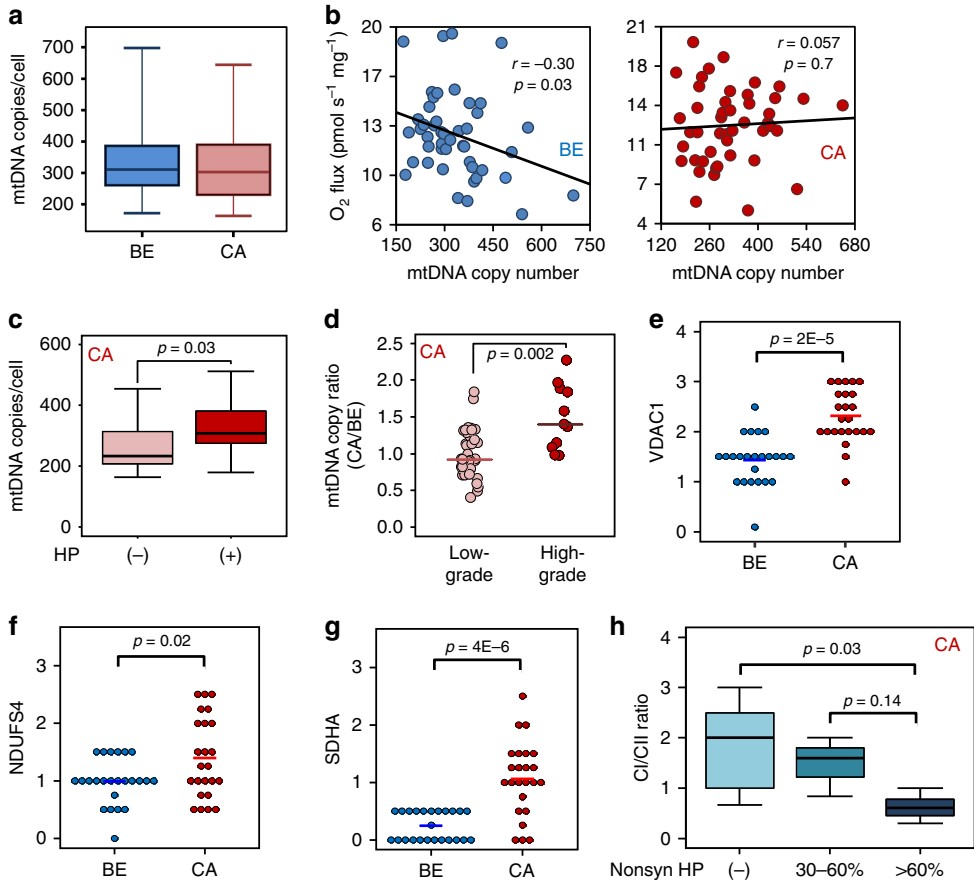

**Fig. 6 mtDNA copy number and mt-mass load. a** Boxplot of mt-CN per cell in the benign (blue, $N = 50$) and malignant (red, $N = 50$) tissue samples. Data are presented as boxplots indicating median, 25th–75th percent percentile (box) and minimum and maximum values (whiskers). **b** Correlation of maximal NS-ET capacity ($NS_E$, $y$-axis) and mt-CN ($x$-axis) in benign (BE, blue circles, $N = 50$) and malignant tissue (CA, red circles, $N = 50$). Linear correlation was established using Pearson's correlation coefficient (r) while correlation was tested for significance using two-tailed $t$-test. **c** Mean mt-CNs ± SD in malignant samples without ((−), light red, $N = 17$) or with ((+), dark red, $N = 33$) mtDNA mutations, respectively. Data are presented as boxplots indicating median, 25th-75th percent percentile (box) and minimum and maximum values (whiskers). Differences in mean values were tested for significance using Wilcoxon rank-sum test. **d** Comparison of mt-CN ratios calculated as mt-CN$_{CA}$/mt-CN$_{BE}$ in tumor samples in low grade (Gleason score ≤7, light red, $N = 40$) vs high grade (Gleason score >7, dark red, $N = 10$) tumors. Differences in mean values were tested for significance using Wilcoxon rank-sum test. **e–g** Tissue immunostaining of mitochondrial markers for mt-mass (VDAC1, **e**), CI (NDUFS4, **f**) and CII (SDHA, **g**) in paired BE and CA tissues, $N = 24$. Quick scores are presented and differences in mean values were tested for significance using Wilcoxon rank-sum test. **h** CI/CII marker ratio, stratified according to mt-CI gene mutation load. No HPs in CI genes, (light blue), HP levels of 30-60% (blue) and >60% (dark blue) in CI genes. Data are presented as boxplot indicating median, 25th–75th percent percentile (box) and minimum and maximum values (whiskers). Differences in mean values were tested for significance using one-way ANOVA followed by Tukey's HSD test. Source data are provided as a Source Data file.

in the high-grade (Gleason score >7) were significantly increased compared to low-grade (Gleason score ≤7) tumors ($p = 0.002$; Wilcoxon rank-sum test, Fig. 6d). This suggests an increase of mtDNA load with higher malignancy and potentially deleterious mtDNA mutations.

To assess differences in mt-mass 24 cases showing varying N-pathway respiratory capacities were analyzed by immunohistochemistry. Consecutive tissue sections were immunostained with antibodies directed to the mt-mass marker porin (VDAC1, voltage-dependent anion channel 1), the CI marker NADH dehydrogenase (ubiquinone) iron-sulfur protein 4, (NDUFS4), or the CII marker succinate dehydrogenase complex subunit A (SDHA). Compared to the corresponding benign tissues all immunomarkers were significantly increased in the tumors (Fig. 6e–g, Supplementary Fig. 5). In tumors harboring high HP level non-synonymous mutations in CI genes the CI/CII marker ratios were significantly decreased whereas mt-mass did not differ (Fig. 6h, Supplementary Fig. 6a-b).

**Transcriptome mirrors altered OXPHOS capacity.** Quantitative mRNA expression analysis via NGS was performed for 16 representative paired tumor/benign samples selected to mirror the HRR properties of the whole sample cohort (Supplementary Table 4, Fig. 7a, compare to Fig. 2g). We analyzed 1158 nuclear genes related to mt-function and metabolism based on MitoCarta 2.0[30]. Comparison of expression of these genes between benign and malignant samples identified 512 differentially expressed genes (false discovery rate, $FDR < 0.01$, Fig. 7b). A Pathway Over-Representation Analysis using the InnateDB Pathway online tool[31] allocated differentially expressed genes to amino acid and pyruvate metabolism, TCA cycle, OXPHOS and fatty acid turnover pathways (Fig. 7c). Expression of key genes related to OXPHOS, TCA-driven pyruvate degradation and succinate provision, and of enzymes involved in ROS detoxification were upregulated in the malignant samples (Fig. 7d). Mt-pyruvate carrier 2 (*MPC2*) and subunits of pyruvate dehydrogenase (*PDHX, PDK1,* and *PDP2*) drive mitochondrial allocation of

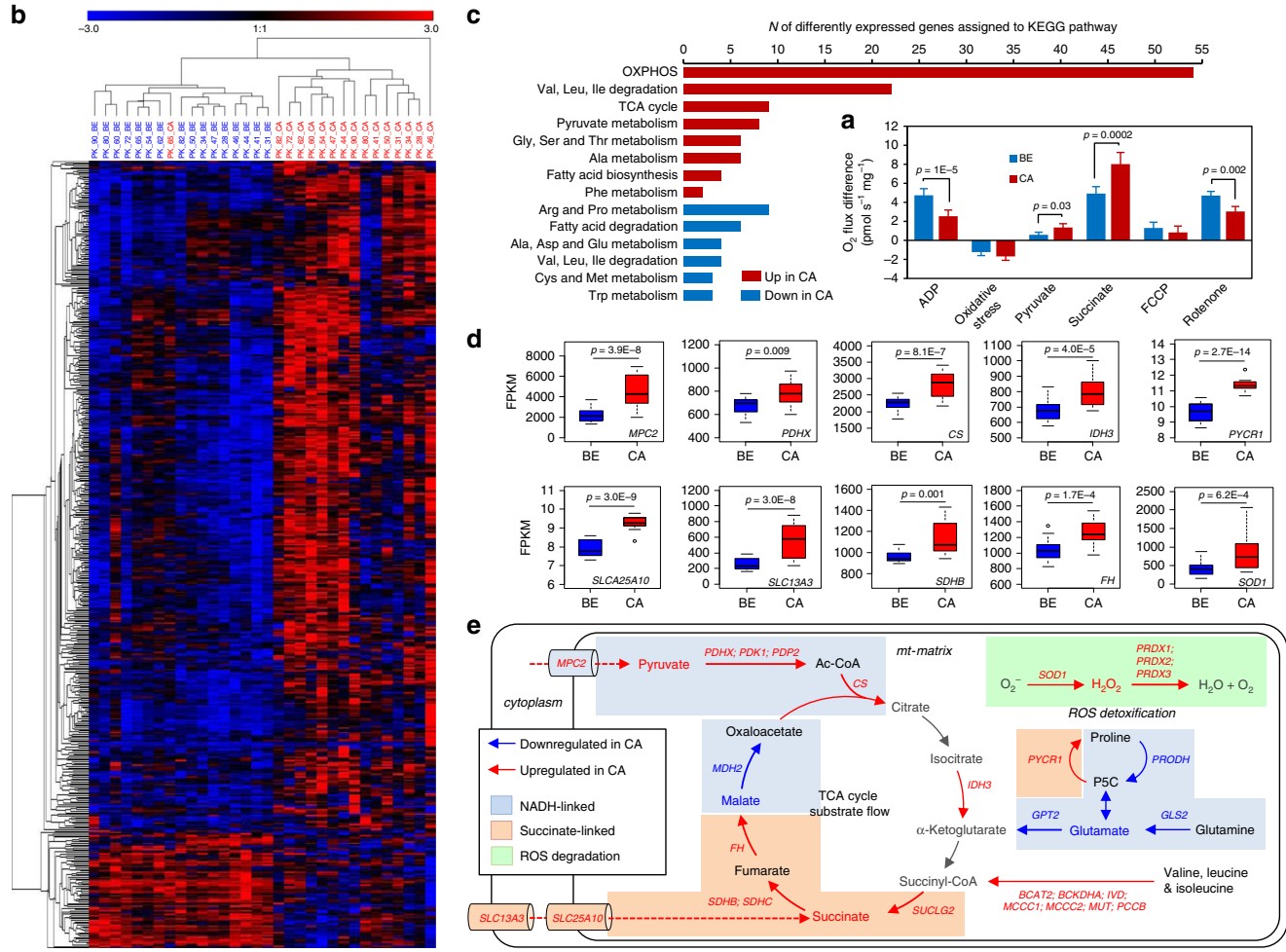

**Fig. 7 Transcriptome profile of mt-related genes.** Differential mRNA expression of mitochondrial genes and enriched metabolic KEGG pathways were analyzed in 16 paired tissue samples representative for the sample cohort. **a** Respiratory characteristics of tissue samples for which RNA-seq expression analysis was performed. The effects of the substrates glutamate&malate, pyruvate, succinate, of oxidative stress, uncoupler FCCP, and CI inhibitor rontenone on $O_2$ flux in benign (blue, $N = 16$) and malignant (red, $N = 16$) tissue samples are shown. Data are presented as mean values ± SD. Differences were tested for significance using Wilcoxon signed-rank test. **b** Heatmap and hierarchical clustering of all significantly differentially expressed mt-related genes (FDR < 0.01) based on the MitoCarta 2.0 Gene Catalog in the benign (blue) or malignant (red) tissue samples. **c** Enriched metabolic KEGG pathways upregulated (red) or downregulated (blue) in cancer tissue based on the InnateDB pathway overrepresentation analysis of all differentially expressed mt-related genes. **d** Boxplots representing Fragments Per Kilobase Million (FPKM) values for expression of significantly increased metabolic key-enzymes in the malignant (red, $N = 16$) vs the benign (blue, $N = 16$) tissue samples. Data are presented as boxplots with median, 25th–75th percent percentile (box), minimum and median ± 1.5 IQR (whiskers) and minimum and maximum values (dots). Differences were tested for significance using multiple $t$-test followed by Benjamini–Hochberg correction for multiple testing. **e** Detailed annotation of single upregulated TCA-cycle key-enzymes in the benign (blue arrows) and malignant (red arrows) samples. Blue boxes mark steps mainly involved in NADH-linked electron transfer, orange boxes highlight steps related to succinate-linked electron transfer, and green boxes mark steps involved in ROS detoxification. Substrates used during the respirometry experiments are indicated in bold type while blue or red colored names and arrows mark enzymes and enzymatic reactions that are higher in either benign or malignant samples, respectively. Full names of enzymes are given in the main text. Source data are provided as a Source Data file.

pyruvate and conversion into lactate or acetyl-CoA (Ac-CoA). Citrate synthase (CS) and subunits of mt-isocitrate dehydrogenase (IDH3) promote conversion of Ac-CoA to citrate and ultimately TCA-driven oxidation of pyruvate into α-ketoglutarate. Mt-(SLC25A10) and cytoplasma membrane (SLC13A3) dicarboxylate transporters, two succinate dehydrogenase subunits (SDHB, SDHC) along with subunits of key TCA-enzymes (succinyl-CoA synthethase SUCLG2, fumarat hydratase FH) and enzymes involved in the conversion of various amino acids into succinyl-CoA (BCAT2, BCKDHA, IVD, MCCC1, MCCC2, MUT, and PCCB) support a more efficient provision of succinate for respiration in tumors. Increased expression of pyrroline-5-carboxylate reductase 1 (PYCR1) enables recycling of proline using NADPH and sustained feeding

of electrons into CII by oxidation of proline by PRODH[32]. In a breast cancer model, this PRODH-PYCR1 cycle was shown to enhance invasion and metastasizing properties of tumor cells[33]. Furthermore, the high expression of genes related to ROS degradation such as superoxide-dismutase (SOD1) and peroxireductin 1-3 (PRDX1, PRDX2, PRDX3), indicates a higher ROS exposure and consequently increased necessity to detoxify oxygen radicals in the tumors.

Genes of enzymes related to glutamine-driven ATP generation via the TCA-cycle (glutaminase, GLS2; glutamate-pyruvate transaminase, GPT2) and proline dehydrogenase (PRODH) involved in the metabolization of proline to glutamate were downregulated in tumors. This is in line with the lower GM-pathway capacity in malignant compared to benign tissue. Genes

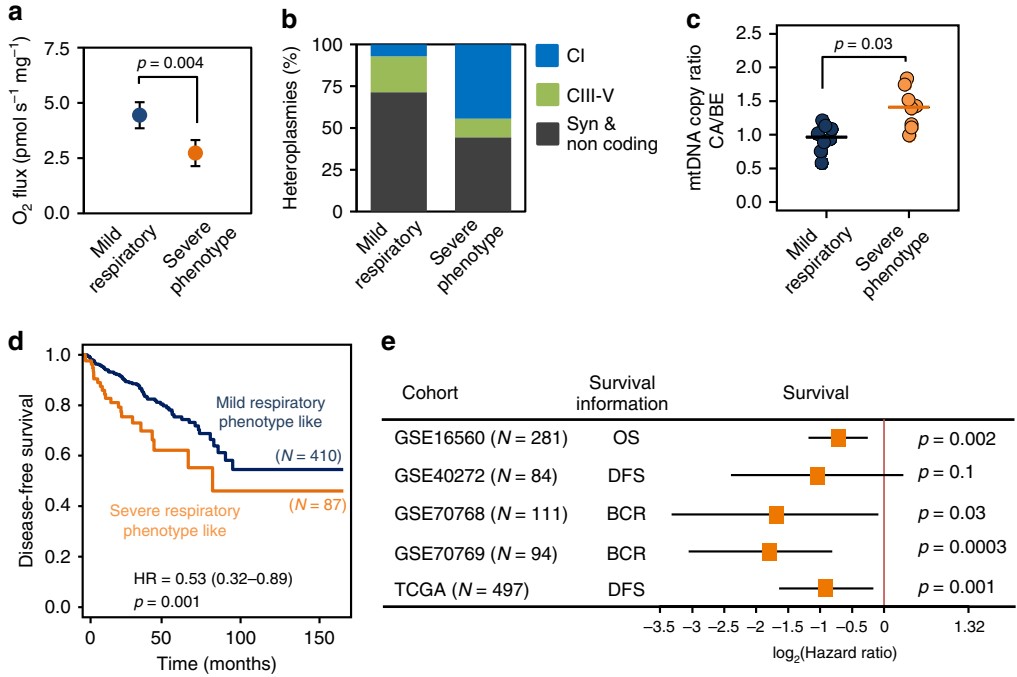

**Fig. 8 Association of severe respiratory phenotype and survival. a** Glutamate&malate-linked OXPHOS capacity ($GM_P$) in the severe (orange) versus mild (blue) respiratory phenotype PCa tissue samples. Data are presented as mean values ± SD. Differences were tested for significance using Wilcoxon rank-sum test. **b** Relative proportion of synonymous (gray), non-synonymous CIII-IV (green) and non-synonymous CI (blue) mtDNA mutations detected in samples of the mild and severe respiratory phenotype tumors, respectively. **c** mt-CN ratios (CA/BE) as determined by duplex qPCR in the severe (orange, $N = 8$) versus mild (blue, $N = 8$) respiratory phenotype PCa samples. Mean and individual data points are presented and differences were tested for significance using Wilcoxon rank-sum test. **d–e** Gene expression profiles and survival information (OS, overall survival; DFS, disease-free survival; BCR, time to biochemical recurrence) were retrieved for five prostate cancer cohorts. The metagene set of 11 genes (Table 2) highly correlated to the gene expression profile of the severe respiratory phenotype PCa samples was used to dichotomize the PCa cohorts into samples exhibiting a mild or a severe respiratory phenotype-like gene expression signature and perform Kaplan–Meier and Hazard ratio (HR) analysis of survival probabilities. **d** Disease-free survival Caplan–Meier curves for the PRAD-TCGA cohort ($N = 497$). A HR of 0.53 (confidence interval 0.32–0.89, $p = 0.001$, stratified log-rank test) was calculated for this set. **e** Survival probabilities (HRs and confidence intervals, CI), for all five independent prostate cancer cohorts ($N = 1067$). Cases exhibiting a severe respiratory phenotype-like metagene signature showed a significantly shorter survival probability in four of the five cohorts. HRs were tested using stratified log-rank tests. Source data are provided as a Source Data file.

involved in regulation of the pyruvate dehydrogenase super-complex (PDH), including pyruvate dehydrogenase kinases (*PDK2, PDK3,* and *PDK4*), were expressed higher in benign tissue suggesting a more stringent regulation of pyruvate catabolism. Taken together, the pathway alterations suggest a consistent pattern distinguishing benign and cancer tissue (Fig. 7e).

**Shorter survival with a severe mitochondrial phenotype.** We performed survival analysis using publicly available PCa cohort data annotated with progression-free or overall survival data to evaluate the prognostic value of the specific expression pattern associated with a decrease in N-pathway respiratory capacity. We grouped our cancer cases with expression profiles into two equal cohorts according to their GM-OXPHOS capacity, a low $GM_P$ ("severe") and a high $GM_P$ ("mild") mt-respiratory phenotype group. The relative GM-respiratory capacity of the severe respiratory phenotype tumors was about half compared to the mild respiratory phenotype tumors ($p = 0.002$, Wilcoxon rank-sum test, Fig. 8a). The severe phenotype group featured a significantly higher incidence of non-synonymous mtDNA mutations, particularly in CI genes (odds ratio=10.4, $p = 0.04$; Fisher's exact test, Fig. 8b). Of note, five severe but only one mild respiratory phenotype tumors harbored a non-synonymous CI gene mutation. The variant level in this single mild phenotype

tumor was only 3.8% compared to 5.4–60.2% in the severe phenotype tumors (Supplementary Table 4). Another distinctive property of the two groups was a significantly higher mt-CN load in the severe phenotype tumors ($p = 0.02$, Wilcoxon rank-sum test, Fig. 8c).

Comparison of the transcriptomes of the severe and mild respiratory phenotype tumors revealed upregulation of branched-chain amino acids and fatty acid degradation, pyruvate metabolism, oxidative phosphorylation and TCA cycle pathways in the severe respiratory phenotype tumors. In contrast the pathways for metabolization of other amino acids (glutamine, alanine asparagine, phenylalanine, arginine, proline) were higher expressed in the mild respiratory phenotype tumors (Supplementary Fig. 7). We used the transcriptome profiles to extract a metagene set of 11 genes strongly correlated to the severe respiratory phenotype tumors, to test if these two distinct phenotypes were associated with different disease outcomes ($r > 0.4$, $p < 0.05$, multiple $t$-test followed by Benjamini–Hochberg correction for multiple testing, Table 2). We then dichotomized samples of the cancer genome atlas (TCGA-PRAD) cohort, $N = 497$) into two cohorts according to their metagene set expression scores using an optimum cut-off value. The high score tumor group exhibited a statistically significantly shorter disease-free survival probability (Fig. 8d; hazard ratio=0.53, confidence interval=0.32–0.89, $p = 0.001$, stratified log-rank test). This result was confirmed with four PCa gene array expression cohorts

**Table 2 Metagene signature of severe respiratory phenotype tumors.**

**Severe respiratory phenotype metagenes**

| Gene ID | Gene name |
|---------|-----------|
| ACADL | Acyl-CoA dehydrogenase long chain |
| ALDH7A1 | Aldehyde dehydrogenase 7 family member A1 |
| AUH | AU RNA binding methylglutaconyl-CoA hydratase |
| BPHL | Biphenyl hydrolase like |
| CHDH | Choline dehydrogenase |
| CRLS1 | Cardiolipin synthase 1 |
| FECH | Ferrochelatase |
| LDHD | Lactate dehydrogenase D |
| MAOA | Monoamine oxidase A |
| MIPEP | Mitochondrial intermediate peptidase |
| NUDT8 | Nudix hydrolase 8 |

Significantly overexpressed genes exhibiting a significant correlation ($r > 0.4$, $p < 0.05$, multiple $t$-test followed by Benjamini–Hochberg correction for multiple testing) were extracted from the "severe" phenotype samples and classified as representative "metagenes" for this phenotype.

($N = 570$) (Fig. 8e, Supplementary Table 5). Together, these results indicate worse outcome associated with metabolically remodeled PCa.

## Discussion

The specific metabolism of prostatic epithelial cells is characterized by secretion of citrate into the prostatic fluid. Exploiting glutamate and other amino acids as anaplerotic fuel substrates and hence as major sources for ATP production allows sparing of pyruvate for citrate synthesis and excretion[14]. In PCa cancer cells, the downregulation of citrate excretion[34] enables the utilization of citrate to cover energy demands. Supporting this hypothesis, benign human prostate tissue in our study showed a significantly higher glutamate&malate-driven OXPHOS capacity while malignant tissue showed an increased utilization of mainly succinate and to a lesser extent also pyruvate to drive energy production and compensation for the reduced N-pathway capacity, in agreement with PCa cell line results[33]. A recent study of OXPHOS activity in PCa cell models demonstrated the importance of especially pyruvate to sustain malignant growth[35]. The shift in substrate preference was mainly present in high-grade tumors, indicating that this metabolic rewiring is a definitive step during tumorigenesis with prognostic significance. It allows sparing of bioprecursors for other needs while a high succinate oxidation is maintained for efficient ATP production. These observations underscore the significance of succinate as an "oncometabolite"[36]. In the light of these findings, succinate and CII are promising targets for novel anti-cancer drugs[33,37].

Sequencing of the mtDNA of our 50 paired samples identified 84 specific HP sites in the tumors and 33 in benign samples, corresponding to a tumor mtDNA HP rate of 1.6 versus 0.6 in the benign tissues. Recent studies of mtDNA NGS reported mutation frequencies ranging from 0.7 to 2.1 per tumor[9–12,20] (Supplementary Table 6). The landscape study of Hopkins et al., which included samples of 384 PCa patients, reported a rate of 0.8. The mtDNA variants identified herein show little overlap with published mtDNA variants in tumors supporting random mtDNA mutagenesis by replication and DNA repair errors and lack of driver mutations as pointed out by Ju et al.[9]. Studies on mtDNA mutations in PCa differ significantly regarding control samples (blood or benign prostate tissue) sample isolation (bulk or microdissected tissue), variant calling and thresholds, thus limiting a direct comparison. Using the threshold of Hopkins et al. ($\Delta > 0.2$) our tumor HP rate is similar (0.5 vs 0.8) whereas our

control rate is half of Hopkins et al. (0.1 vs. 0.2) (Supplementary Fig. 8). It is important to consider the different matched control tissues (paired benign vs. blood tissue) as mtDNA mutations vary across organs, even within one individual[38,39].

Our study highlights non-synonymous, potentially deleterious mutations in mtDNA-encoded genes of the mitochondrial machinery as the crucial alterations impacting on mt-function in primary PCa. Non-synonymous mutations in mt-CI protein genes turned out to be closely linked to respiratory and metabolic differences, whereas mutations in genes encoding proteins of other mt-complexes seem to be better tolerated. This warrants exploration of of mtDNA CI HPs as diagnostic indicators of PCa progression risk.

mtDNA heteroplasmy levels up to 80% may exhibit little impact on energy production[40]. Although this observation is well supported by deleterious mtDNA mutations affecting CIV, ATP synthase or tRNAs, like the *A3243G* tRNA$^{Leu}$ mutation causing MELAS syndrome or the *A8344G* tRNA$^{Lys}$ mutation causing MERRF syndrome[41,42], CI gene mutations lead to symptomatic mt-myopathies at much lower thresholds. LHON or Leigh's syndrome patients consistently show ~50% decrease in N-pathway capacity at variant levels of 20-50%[24,25,43,44]. These findings are in agreement with our study with heteroplasmy levels of 30–60% exhibiting a significant N→S-pathway shift. Strong support for a causal link between potentially deleterious mtDNA mutations and the respiratory alterations comes from: (1) strong N→S-capacity phenotype in samples harboring CI gene mutations, (2) clustering of these mutations to the central axis of CI, a highly-charged channel involved in redox coupling and proton translocation[27], (3) significant structural impact by mutations indicated by 3D modeling, (4) increased mtDNA content and (5) increased mt-mass marker and reduced CI/CII maker ratio by IHC in tumors with high mutation load in mt-CI genes. The mt-IHC marker pattern confirm upregulation of CII as an important mechanism to enable sustained OXPHOS capacity for ATP production in tumors despite CI protein mutations affecting CI and N-pathway contribution. Upregulation of succinate-triggered OXPHOS respiration upon inhibition of N-pathway oxidative flux in prostate cells indicates that the S-pathway is in general not working at its maximum capacity. This allows to recruit further S-capacity to keep OXPHOS high when oxidative flow from CI to the Q junction is compromised, e.g., by CI gene mutations.

Importantly, our results show that primary human PCa tissue is highly capable of aerobic ATP production. Despite decreased N-pathway capacity, the cancer cells reorganize OXPHOS capacities and metabolism to compensate this loss. A high respiratory capacity is crucial for high malignant and invasive potential[45]. In contrast to tumors, human skeletal muscle and fibroblasts heavily rely on N-pathway capacity[46,47] and a similar compensatory mechanism is not known for mt-myopathies. Ultimately, our results suggest that the immediate net effect of deleterious mtDNA mutations is compensated and does not hamper cellular ATP homeostasis in primary human PCa. Nevertheless, the association with shorter survival and in vitro and in vivo data indicate that deleterious *MT-ND* variants increase the tumorigenic potential[48,49]. This points to a significance of such mutations beyond the immediate effect on cellular energy supply.

Non-synonymous mtDNA mutations are associated with specific changes in the expression of OXPHOS-related proteins[42]. In our study, pathways linked to glutamate metabolism were decreased, those related to pyruvate and succinate utilization increased in tumor tissue, thus mirroring the respiratory patterns and suggesting a specific significance of such mutations to rewire metabolism. The mechanistic link between mutations and altered enzyme expression levels remains elusive at this stage, however, our results provide a first view on altered expression patterns that

can guide future studies of tumorigenesis and progression-supporting metabolic pathways.

Metabolic alterations and high load of non-synonymous mtDNA mutations are accompanied by an increase in mtDNA content and mt-density, which is a characteristic of more aggressive PCa[21,50]. Confirming this hypothesis, a set of highly correlated metagenes extracted from the gene expression profile of PCa samples exhibiting a severe respiratory phenotype was able to predict significantly shorter survival. Taken together, the severe N→S-shifted respiratory phenotype provides an additional prognostic indicator for aggressive PCa.

Our study has limitations: (1) functional changes observed by HRR analysis were detected using artificially high concentrations of substrates and oxygen, which may lead to an overestimation of the effects. However, tissues are composed of a mixture of malignant and non-malignant cells (e.g., the stroma), resulting in a "dilution" of the malignant phenotype. (2) HRR analysis involves multiple sequential enzymatic steps and further experiments exploring isolated TCA cycle proteins and/or ETS subunits are needed to evaluate their single contribution in more detail. On the other hand, our method allows a realistic evaluation of the complex interplay of the various enzymatic steps of respiration. (3) We focused on evaluation of functional and genomic data, while gene expression was analyzed at mRNA level and protein levels by IHC for mt-markers only. It is widely accepted that the combination of HRR and mtDNA sequence analysis provides the most powerful tool to study the effect of (potentially) deleterious mtDNA variants on TCA cycle and OXPHOS function[51–53]. The IHC expression data of mt-markers and a recent immunohistochemical study revealed a significantly elevated expression of CII-protein subunits in primary PCa tissue[54]. In line with that, Grupp et al.[55] reported increased mitochondrial mass associated with PCa progression. (4) More comprehensive functional characterizations of the potentially deleterious mtDNA variants described in our study are necessary to evaluate their impact on N-pathway function and cellular ATP homeostasis. Patient follow-up is warranted to estimate the effects on long-term disease outcome.

In conclusion, this is the first study analyzing substrate-specific OXPHOS capacities in native primary human prostate tissue. It links remodeling of OXPHOS in PCa to potentially damaging mtDNA mutations and differential expression of mt-genes. Our findings suggest that decreased N-pathway capacity associated with potentially deleterious, high-level mtDNA heteroplasmies in mt-CI genes, higher mtDNA load and increased mt-mass are distinct characteristics of high-grade tumors, highlighting the diagnostic and prognostic potential of metabolic rewiring. Analysis of the CI molecular structure suggested a structural basis for the effects of functionally relevant mutations. Furthermore, we provide evidence that both, potentially deleterious mtDNA mutations and an aberrant gene expression account for these alterations in a subgroup of PCa samples. We defined a severe respiratory phenotype PCa subtype characterized by a significant N→S-pathway shift and show that the distinct expression signature of this subgroup is associated with worse disease prognosis in large and independent PCa cohorts. This signature could thus help to identify patients at a higher risk, complementing the classical tumor grading and risk assessment system. Our results warrant exploring therapeutic strategies to target alterations of metabolism in PCa and particularly suggest succinate-linked respiration as a therapeutic target.

## Materials and methods

**Prostate cancer subjects**. Fifty prostate cancer patients were recruited for the study and radical prostatectomy was the first-line treatment in all cases (Table 1). All patients underwent surgery at the University Hospital for Urology of the Medical University of Innsbruck and received standard of care radical

prostatectomy. Immediately after surgery, prostate specimens were transported to the Department of Pathology where small tissue samples were excised for the study. Written informed consent was obtained from all patients. The use of human prostate tissue for this study, which was conducted in accordance with the principles of the Declaration of Helsinki, was approved by the Ethics Committee of the Medical University of Innsbruck (AN 4837). Clinical characteristics are summarized in Table 1.

**Cell cultures**. LNCaP and PC3 prostate cancer and benign prostate epithelial RWPE1 cell lines were purchased from the American Type Culture Collection (ATCC; Rockville, MD). DuCaP prostate cancer cells were a gift from Dr. Schalken (Radboud University, Nijmegen, The Netherlands). The human benign prostate epithelial EP156T cell line was established by overexpression of hTERT[56]. Tumor cells were grown in RPMI 1640 medium (Szabo-Scandic, Vienna, Austria) containing 10% fetal bovine serum (FBS; PAN BioTech, Aidenbach, Germany), glutamax (Thermo Fisher Scientific, Vienna, Austria) and antibiotics (Szabo-Scandic). Benign cells were grown in RPMI 1640 medium containing 5% FBS, glutamax, 0.5x ITS (insulin-transferine-sodium selenium, Thermo Fisher Scientific), 12.5 mg/mL bovine pituitary extract (Thermo Fisher Scientific), 2 ng/mL EGF, 10 nM testosterone (Sigma), 100 nM hydrocortisone (Sigma) and antibiotics. All cell lines were maintained at 37 °C in a humidified atmosphere with 5% $CO_2$.

**Tissue samples and diagnostic confirmation**. Tissue biopsies were extracted from prostate specimens by an experienced uropathologist using punch needles. For each individual, paired tissue samples consisting of one malignant and one non-malignant benign biopsy were collected (Fig. 1). A small portion of the biopsy was formalin-fixed, dehydrated and paraffin-embedded for subsequent diagnostic staining and validation of tissue identity, while the remaining tissue biopsy was placed into pre-chilled relaxing and preservation solution BIOPS, containing 2.77 mM $CaK_2EGTA$, 7.23 mM $K_2EGTA$, 20 mM imidazole, 20 mM taurine, 50 mM MES hydrate, 0.5 mM DTT, 6.56 mM $MgCl_2$, 5.77 mM ATP and 15 mM phosphocreatine. Diagnostic eosin-hematoxylin (H&E) and basal cell marker p63/tumor cell marker AMACR double-immunostaining (IHC) was performed for each biopsy and the respective surrounding area for confirmation of tissue identity (Fig. 1b). IHC was performed with 5 µm tissue sections employing the Ventana Discovery-XT instrument (Roche, Vienna, Austria). Standard CC1 pre-treatment for antigen retrieval was followed by incubation with antibodies diluted in antibody diluent (AMACR-P405S, DAKO M361601-2, dilution 1:400 and P63-4A4, Roche 05867061001, dilution 1:4), secondary universal antibody solution for 30 min, staining with DAP map kit and counter staining for 4 min with hematoxylin II bluing reagent (all reagents from Roche). Specificity of staining was controlled by including a control antibody (DAKO).

**Reagents**. Stock solutions for the respirometry measurements were stored frozen except for the $H_2O_2$ titration solutions, which were prepared freshly on a daily basis. High-performance liquid chromatography (HPLC) or polyacrylamide gel electrophoresis (PAGE) grade primers, probes and barcode oligonucleotides were ordered from Microsynth (Microsynth AG, Balgach, Switzerland) as 100 µM stock solutions, stored frozen and freshly diluted prior to use.

**High-resolution respirometry (HRR)**. An established substrate-uncoupler-inhibitor-titration (SUIT) HRR protocol for prostate tissue samples[14] was modified for analysis of metabolic and respiratory capacities of paired tissue samples used in our study (SUIT-028, [http://www.mitofit.org/index.php/SUIT-028]). Mechanical permeabilization of the tissue samples was performed in a small glass petri dish on a pre-chilled metal plate in MiR05 + creatine (3 mg/mL, MiR05Cr) buffer with two pairs of sharp forceps for 5 min as reported previously[14]. Each permeabilized tissue sample was divided into two smaller samples of equal size, rinsed with ice-cold MiR05Cr, briefly blotted on a filter paper before wet tissue mass was determined. Following permeabilization, each tissue sample was placed into a pre-calibrated chamber of an Oxygraph-2k (O2k, Oroboros Instruments, Innsbruck, Austria). Split benign and malignant biopsies were analyzed simultaneously (duplicate measurements for each sample type in four chambers). All experiments were performed at 37 °C under constant stirring while oxygen concentration was kept between 200 and 300 µM. For real-time data acquisition and post-experimental analysis, DatLab software (V6 and V7.4, Oroboros Instruments) was used. Substrate concentrations, their primary sites of action and the corresponding coupling/pathway control states are summarized in Supplementary Tables 1–2[57]. Respiratory capacities were expressed as oxygen consumption per wet mass of tissue (pmol s$^{-1}$ mg$^{-1}$) and corrected for residual oxygen consumption (ROX), measured after inhibition of Complexes CI, CII, and CIII. Mean values of split samples were used for statistical analysis. For determination of relative respiratory capacities (flux control ratio, $FCR$) respiratory capacities were normalized to the internal reference state $NS_E$.

We applied SUIT-028 protocol to evaluate the activity of different segments of the electron transfer system in three distinct coupling/pathway control states, LEAK, OXPHOS and ET (Fig. 2a–c). LEAK respiration was measured in the presence of the NADH-linked substrates glutamate (10 mM) and malate (2 mM) (GM), which transfer electrons along CI into the Q-pool. For determination of the

corresponding OXPHOS capacity, ADP (2.5 mM) was added at saturating concentrations. This was followed by incubation with $H_2O_2$ (500 μM) for 15 min in the closed chamber to induce oxidative stress and assess the effect of damage mediated by elevated cellular ROS levels[58]. The excess $H_2O_2$ was then removed by addition of catalase (280 U/mL) and the chambers were briefly opened to regain initial $O_2$ concentrations. After titration of pyruvate (P, 5 mM), succinate (S, 10 mM) was added to fuel electrons via CII, thus reconstituting TCA cycle function and inducing convergent NADH- and succinate-linked (NS) electron flow through CI and CII into the Q-juction. Subsequently stepwise titration of uncoupler carbonyl cyanide p-trifluoro-methoxyphenyl hydrazone (FCCP, 0.5 μM) enabled the determination of maximal non-coupled NS-respiratory capacity. Inhibition of CI by rotenone (Rot, 0.5 μM) revealed S-pathway ET-respiratory capacity. Residual oxygen consumption (Rox) was measured after additional inhibition of CII and CIII by malonate (Mna, 5 mM) and antimycin A (Ama, 2.5 μM), respectively.

In control experiments assessing the effect of simulated oxidative stress by $H_2O_2$ exposure, tissue samples were split, one half was analyzed using the $H_2O_2$ treatment step, while the other half was incubated with $H_2O$ as a control. In addition, a series of paired BE/CA tissue samples was measured using a protocol omitting the $H_2O_2$ treatment. The impact of oxidative stress treatment on the single enzymatic step of Complex IV was assessed in split BE tissue samples using a protocol including inhibition of CI, CII, and CIII with Rot, Mna and Ama. Ascorbate was added followed by N,N,N',N'-Tetramethyl-1,4-phenylenediamine (TMPD). Chemical background oxygen consumption was measured after inhibition of CIV with sodium azide and was used for correction (Supplementary Tables 1–2).

For HRR measurements in cell lines, cells were harvested by trypsinization, counted and re-suspended in MiR05 buffer. Measurements were performed with 0.5–1 Mill cells per mL in the O2k chamber and the SUIT protocol used for tissue samples with an additional step of plasma membrane permeabilization by titration of digitonin was applied (SUIT-014, [http://www.mitofit.org/index.php/SUIT-014]). To simulate injury of CI, rotenone was added in stepwise titration of 1 nM until partial inhibition of N-pathway capacity in OXPHOS. In control experiments vehicle (ethanol) was added.

**Next-generation sequencing of the entire mitochondrial genome.** After the HRR experiments tissue biopsies were recovered, washed two times with pre-chilled PBS and stored at −80 °C until isolation of DNA. Total genomic DNA was extracted using the EZ1 DNA Tissue Kit (Qiagen, Hilden, Germany) following the manufacturers' instructions. The whole mt-genome was amplified in two overlapping PCR fragments of app. 8.5 kbp using the primers 5′-AAATCTTACCC CGCCTGTTT-3′ (fragment A, forward), 5′-AATTAGGCTGTGGGTGGTTG-3′ (fragment A, reverse), 5′- GGCAGGTCAATTTCACTGGT-3′ (fragment B, forward), 5′- GCCATACTAGTCTTTGCCGC-3′ (fragment B, reverse)[23,59]. After PCR, all amplicons were quality checked after purification on a Fragment Analyzer using the DNF-930 dsDNA Reagent Kit (both Advanced Analytical, Ames, IA).

For generation of barcoded 200 bp insert Ion Torrent NGS libraries, 1 μg of DNA of a 1:1 mixture of both mtDNA fragments A and B was enzymatically fragmented using the NEBNext dsDNA Fragmentase Kit (New England Biolabs, Ipswich, USA), end-repaired with the NEBNext End Repair Module (New England Biolabs) and ligated with IonXpress adapters (Life Technologies, Waltham, MA) using T4 DNA Ligase and Bst DNA Polymerase (New England Biolabs). The ligation products were size selected on an E-Gel SizeSelect Agarose Gel, and amplified using the NEBNext High-Fidelity Kit (New England Biolabs). All post-reaction purification steps were conducted using MagSi-NGS$^{PREP}$ magnetic beads (Magna Medics Diagnostics, Geleen, NL). Size distribution of the enriched libraries was assessed on Fragment Analyzer while the concentration was determined on a 7900HT Fast Real-Time PCR System. Template positive Ion Sphere Particle (ISP) generation and enrichment, followed by automated chip loading were performed on the Ion Chef System (Life Technologies) using the Ion PI IC 200 Kit (Life Technologies). Sequencing on the Ion Proton Sequencer was performed using Ion PI Chips and Ion PI Sequencing 200 Kit v2 chemistry (Life Technologies). Each run included artificial control NGS libraries prepared by mixing the mtDNA of two individuals in a 1 + 9, 1 + 49 and 1 + 99 ratio to generate libraries with known and well-defined HP levels. Each library was sequenced twice via a total of three different runs on three different plates on the Ion Torrent Proton, comprising controls (such as predefined sample mixtures of two previously known haplotypes, and internal control samples). Subsequently the data were joined, by only considering heteroplasmic and homoplasmic variants occurring in two different sequencing runs. Heteroplasmic levels are represented as the mean variant allele frequency of a mutation in the two runs. Concordance between the sequencing runs was confirmed by an analysis of the detected variants via the Bland-Altman plot (Supplementary Fig. 3).

**Analysis of mtDNA next-generation sequencing data.** mtDNA data were analyzed using an in-house installation of mtDNA-Server[60]. Sequence reads were aligned to the revised Cambridge Reference Sequence (rCRS)[61] with a distributed version of BWA MEM 7.5 based on JBWA (mtDNA-Server, [https://mtdna-server.uibk.ac.at/]). Haplogroups were determined via HaploGrep[62] to also exclude potential sample contaminations. The data were processed with the following

parameters: Phred Quality score per base filtering was set to Q20 (1% error rate), alignment score of Q30, mapping Quality Score 20, and the per base alignment quality (BAQ) filtering was applied by adapting GATKs implementation of BAQ for circular genomes. Only HPs exceeding a 2% threshold were accepted for analysis in order to avoid false positive results. This threshold was determined based on the precision and recall accuracy of the four control samples (1 + 1, 1 + 9, 1 + 49 and 1 + 99 mixtures of two individuals[60]. A minimal coverage per strand of >10 bases was considered, while a minimal coverage of 5 bases per strand was required for detection of heteroplasmy variants. Forward and reverse strand were required to show similar results, verified by the strand bias. Private mutations were defined as HPs found in one tissue type only, be it cancer or benign, as opposed to shared HPs present in both tissue types. All found HPs are listed in Supplementary Data 1, categorized in "private benign", "private cancer" and "shared" HPs.

**Pathogenic effects of mt-mutations on protein function.** To compare mutation profiles in the benign-cancer tissue pairs, variants where classified into three categories: benign variants only, shared variants (assumed as germline variants) and tumor variants only. The variants were subsequently annotated with the pre-computed MutPred scores as provided by Pereira et al in Supplementary Table 3 of their article[19] and the pre-computed pathogenicity predictions provided by MitImpact 2[15,63] by downloading the data as tab-delimited file and merging in a relational database.

**mt-Complex I protein structure modeling.** Protein structures were inspected and analyzed using COOT 0.8.7.1[64]. Mutations were placed using the mutation function in COOT. The ND domains of Thermus thermophilus CI crystal structure (PDB: 4HE8 and 4HEA, [https://www.rcsb.org/])[27] were superimposed with the cryo-EM model of the CI subunit of the human respiratory megacomplex (PDB: 5XTD and 5XTC)[26] in COOT in order to relate functional details described for T. thermophilus to the human cryo-EM structure of CI. Structure figures were generated and rendered with Chimera 1.1.2[65]. A supplemental iSee package[66] has been produced to visualize and examine the functionally relevant mutation sites and can be viewed online in any JavaScript enabled web browser ([https://github.com/genepi/mt-c1]).

**mtDNA copy number determination.** Starting from tissue total DNA extracts mtDNA copy numbers (mt-CN) were determined via duplex qPCR by simultaneous quantification of a mitochondrial (MT-TL1) and a nuclear (B2M) amplicon on a Quantstudio 6 instrument (Life Technologies)[43]. For improving assay accuracy and minimizing inter-assay variability a plasmid construct containing the MT-TL1 and the B2M amplicons was included as a reference standard[28]. mtDNA fragment forward primer 5′-CACCCAAGAACAGGGTTTGT, reverse primer 5′-TGGCCATGGGTATGTTGTTA; nDNA forward primer 5′-TGCTGTCTCCA TGTTTGATGTATCT, reverse primer: 5′-TCTCTGCTCCCCACCTCTAAGT, Probe sequences FAM-5′-TTACCGGGCTCTGCCATCT-BHQ1 and Yakima Yellow-5′-CAGGTTGCTCCACAGGTAGCTCTAG-BHQ1, respectively. mt-CN per diploid cell was calculated according to the formula $CN = 2xE^{-\Delta\Delta Cq}$, where "ΔΔCq" denotes $(Cq_{mt} - Cq_n)_{sample} - (Cq_{mt} - Cq_n)_{plasmid}$, allowing for the plasmid-based correction.

**Immunohistochemistry for mt-markers.** IHC staining was performed using the antibodies anti-VADC1 (D73D12, Cell Signaling Technologies, Leiden, The Netherlands; 1:400,), anti-SDHA (D6J9M, Cell Signaling Technologies; 1:400) and anti-Ndufs4 (EP7832, Abcam, Cambridge, UK; 1:200)[54]. Immunostaining was evaluated using a modified "quickscore" procedure considering staining intensity and the percentage of stained cells with scores ranging from 0 to 3 (0 = absent, 1=weak, 2=intermediate, 3=strong). In cases with two recognizable tumor foci or tumor patterns, both foci were scored and mean scores taken for statistical analysis.

**Global transcriptome next-generation sequencing.** Global gene expression analysis was performed by RNA-NGS-sequencing. RNA was isolated from frozen tissue blocks from areas surrounding the tissue samples extracted for HRR/mtDNA sequencing (Fig. 1a). Libraries were prepared using the Illlumina TruSeq Stranded mRNA LT Sample Preparation Kit and paired-end sequencing was performed on an Illumina HiSeq 2500 instrument (Illumina, San Diego, CA) according to the manufacturer's instruction. The mean relative GM-pathway capacity for all malignant samples in this study was 0.27 ± 0.10 as determined by HRR. Therefore, a GM- pathway contribution of ≤27% relative to maximal OXPHOS capacity was considered as "low" whereas a relative GM-pathway contribution of >27% was considered as "high" GM-pathway capacity.

**RNA expression analyses.** As a first step before raw sequence data analyses, the FastQC tool was applied as quality control on the sequencing reads from all datasets. The sequencing reads were aligned to the human genome (GRCh38/hg38 assembly) using STAR and processed with Samtools[67,68]. Gene-level read counts were obtained with htseq-count of the Python package HTSeq, using the default union-counting mode[69]. The differential gene expression analysis of tumor and non-malignant benign samples was calculated using the DESeq2 package in R[70].

Clustering and visualization were done with the software Genesis[71]. A collection of 1158 human mt-genes for expression analysis was retrieved from MitoCarta2.0 database[30] ([https://www.broadinstitute.org/files/shared/metabolism/mitocarta/human.mitocarta2.0.html]).

**Metabolic pathway assignment**. For pathway analysis and annotation the InnateDB Pathway Analysis online tool was used ([http://www.innatedb.ca/redirect.do?go=batchPw])[31]. Overexpressed mt-genes as identified by the RNA-Seq analysis were used to perform a Pathway Over-Representation Analysis. The KEGG DAVID analysis list restricted to the proteins and enzymes listed in the MitoCarta2 list was used to group the differentially expressed mt-related genes into respective pathways. To test for significant differences in gene expression between benign/malignant and mild/severe phenotype samples, a pairwise comparison between groups for each gene listed in the MitoCarta2 catalog was performed based on the FPKM values. To control for type-I error accumulation, $p$-values were corrected by the Benjamini–Hochberg procedure.

**Severe respiratory phenotype metagene signature**. For the identification of a set of differentially expressed genes representative for the severe (low GM) mitochondrial respiratory phenotype ("metagenes"), we selected only significantly correlated genes with an average correlation $r > 0.4$ ($p < 0.05$, multiple $t$-test followed by Benjamini–Hochberg correction for multiple testing). We thus tried to avoid that their correlation might be due to chance while preserving a reasonable number of genes for the severe phenotype metagene set[72]. Clinical variables including disease-free survival (time to biochemical tumor recurrence) and the corresponding RNA-seq expression profiles of prostate adenocarcinoma (PRAD) patients listed in the Cancer Genome Atlas (TCGA) database were downloaded directly from the TCGA Data Portal ([https://cancergenome.nih.gov/]). For the confirmatory meta-analysis, available clinical data and microarray expression profiles from four additional prostate cancer cohort studies were downloaded from the Gene Expression Omnibus ([https://www.ncbi.nlm.nih.gov/geo/]): GSE16560 (Human 6k Transcriptionally Informative Gene Panel for DASL; $N = 281$)[73], GSE40272 (SMD Print_1529 Homo sapiens; $N = 84$)[74], GSE70768 (Illumina HumanHT-12 V4.0 expression beadchip; $N = 111$)[75] and GSE70769 (Illumina HumanHT-12 V4.0 expression beadchip; $N = 94$)[75]. The survival information (overall survival OR, disease-free survival DFS or time to biochemical recurrence BCR, respectively) was extracted from the clinical XML files in the complete clinical sets.

**Statistical analyses**. All statistical analyses were performed using IBM SPSS or R software version 2.0 and numerical data are presented as mean ± SD unless otherwise stated. Normality tests were performed using the Shapiro–Wilk Test followed by Q-Q Plot and histogram evaluation. Accordingly, either Student's paired-samples $t$-test or Wilcoxon signed rank test was used for group comparisons whereas Wilcoxon rank-sum test and student's $t$-test was used to compare variables among unpaired samples. If multiple paired $t$-tests were performed on the same sample set, results were corrected according to the Bonferroni-Holm method. For multiple comparisons, one-way analysis of variance (ANOVA) followed by Tukey's honest significant difference (HSD) test was used. Linear regression models followed by one-way ANOVA were used to test for relationships between independent variables. A multivariate logistic regression analysis was performed to evaluate the relationship between several independent variables (patient age, free PSA ratio and tumor stage) and the presence or absence of heteroplasmies as the dependent variable. Spearman's rank correlation coefficients were calculated to evaluate the presence of significant correlations among non-parametric variables and to test for multicollinearity among single independent variables used in the logistic regression model. The significant differential expression of genes was determined using a false discovery rate ($FDR$) below 0.01 as determined by the Benjamini–Hochberg procedure to control for type-I errors after multiple testing. To test associations between categorical variables either Pearson's Chi-squared test or Fisher's exact test was used. Survival times were defined using the latest information. For survival analysis, the patients were dichotomized based on the expression levels of the low-GM metagenes. The optimal cut-points were searched within the inner 80% selection interval and chosen based on a minimal corrected $p$-value and based on a maximum Harrell's C indices. Survival probability curves were calculated via the Kaplan-Meier method while differences in disease outcome were analyzed using Hazard ratios and stratified log-rank test. Univariate and multivariate Cox regression analysis was performed using age, Gleason score, TNM staging and total serum PSA values as binary covariates, for all cohorts mentioned above. For statistical analyses, IBM SPSS or R software version 2.0 including the survival package were used. $p \le 0.05$ was considered not statistically significant.

**Reporting summary**. Further information on research design is available in the Nature Research Reporting Summary linked to this article.

## Data availability

Data that support the findings of this study are available within the article and its supplementary files or from the authors upon reasonable request. Supplementary Information provides Supplementary Tables and Figures, Supplementary Data 1 provides a list of mtDNA Heteroplasmies, the Source Data File provides datasets underlying Figs. 2–8 and Supplementary Fig. 1-4 and 6-8. The Supplementary Software iSee Package for 3D visualization of Complex I mutations is provided for download or online access at [https://github.com/genepi/mt-c1]. DNA and RNA sequence datasets have been disposed in EGA (European Genome-Phenome Archive [www.ega-archive.org]) under the following accession numbers: EGAD00001005931 (RNAseq data set, [https://ega-archive.org/datasets//EGAD00001005931] and EGAD00001005945 (mtDNA data set. Prostate cancer gene expression datasets with survival information were accessed at the Cancer Genome Atlas (TCGA) Data Portal ([https://cancergenome.nih.gov/]): TCGA-PRAD and the Gene Expression Omnibus portal ([https://www.ncbi.nlm.nih.gov/geo/]): GSE16560, GSE40272, GSE70768 and GSE70769. Protein structure data were accessed at the Protein Data Base (PDB, [https://www.rcsb.org/]: 4HE8, 4HEA, 5XTD ([https://www.rcsb.org/structure/5XTD]), 5XTC.

## Code availability

The iSee package can be accessed from GitHub using the following link https://github.com/genepi/mt-c1

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

## Acknowledgements

The study was funded within the COMET K1 Center Oncotyrol—Center for Personalized Medicine by the Austrian Research Promotion Agency (FFG) and the Tyrolean Future Foundation, and the K-Regio project MitoFit funded by the Tyrolian Government, a contribution to COST Action MitoEAGLE supported by COST (European Cooperation in Science and Technology). We thank Christoph Seifarth and Sarah Peer for excellent technical assistance, Maio Bedek for server administration, Dr. Barbara Kofler, Dr. René Feichtinger, Dr. Iris Eder and Dr. Natalie Sampson for helpful discussions.

## Author contributions

H.K., E.G., and B.S. conceived and designed the research; B.S., G.S., H.W., A.N., L.F., V.B., U.G., E.G., F.K., A.N., and H.K. planned experiments; B.S., G.S., A.N., F.F., A.C.S.A.S. and J.I.G. performed experiments; V.B., I.G., P.S., and U.S. performed RNA-seq; A.N. and B.R. performed structural analyses; all authors analyzed and/or interpreted data; B.S., H.W., F.F., C.P., A.N., E.G. and H.K. wrote the manuscript; H.K. organized and supervised the study. All authors reviewed and approved the manuscript for publication.

## Competing interests

E.G. is founder and CEO, and J.I.G. is employee of Oroboros Instruments, Innsbruck, Austria. All other authors declare no conflict of interests.
