## [Peer Review File · Nature Communications]

Reviewers' comments:

Reviewer #1 (Remarks to the Author):

My major concerns are detailed below. My overriding concern is that the authors may not be correctly defining heteroplasmy. Details on that are given in point 1. If I assume that they are correctly defining heteroplasmy, then the reported heteroplasms are at too small a level to be reasonably causing any functional effect on the mitochondria (point 2).

1) My fundamental concern with this paper is that the observed mtDNA heteroplasms are all at relatively low levels, 3% to <50% with an average value of only about 10%. Since all of the reported heteroplasmy values (unless I missed something) are below 50% I wonder if the authors are defining heteroplasmy correctly, as the frequency of the allele different from the reference sequence, rCRS. The maximum of 50% raises the possibility that the authors are instead defining the heteroplasms as minor allele frequencies, which is not the correct way to define it. A heteroplasmic variation from the rCRS of 90% is completely different from a heteroplasmy of 10%. If the heteroplasmy levels are being reported as minor allele frequencies, then this needs to be changed over to the frequency of differences from the rCRS before a correct interpretation of these heteroplasms can be made.

2) Assuming that the authors are defining the heteroplasmy levels correctly (as differences from the rCRS), then the low heteroplasmy levels they are reporting should not affect mitochondrial function. Experience based on proven pathogenic mtDNA variants has taught us that a rather high threshold of 60-80% heteroplasmy must be passed before mitochondrial function begins to decrease. Indeed, the authors report that the heteroplasms do not affect OXPHOS function, which is consistent with my expectation from heteroplasms below the 60-80% heteroplasmy threshold.

3) The authors report that these heteroplasms affect glutamate-malate pathways. I see no convincing arguments here for why mutations in the electron transport chain should affect those pathways and not the OXPHOS function that they are directly involved in. The model that they lay out in Figure 8 argues for why altered gene expression can lead to the glutamate-malate pathway differences, but does not really address the point about the effect of apparently low-level heteroplasms in the mtDNA on this pathway.

4) Page 20: No definition was given in the methods section of how "private" heteroplasms were defined. I presume this was from a comparison against some public database of reported heteroplasms. This definition can vary greatly depending on the database used. This needs to be included in the methods.

5) Page 24-25: The transcriptome analysis begins by restricting to 858 genes from the MitoCarta catalog. If the analysis is truly restricted to those 858 MitoCarta genes, then KEGG pathway analysis is not valid here since only a subset of genes involved in mitochondrial functions was tested. The authors need to be clear what null hypothesis was used in the KEGG DAVID analysis. The default null hypothesis for this tool would be all genes, which is not correct here. If only the 858 mitocarta gene expressions were analyzed, then the null hypothesis should be limited to those 858 genes, not the full gene list. No details are given in the Methods section for this analysis. If the full human gene list was used in the KEGG DAVID analysis, then I see no reason for mentioning the MitoCarta list.

6) A related point is that the MitoCarta list was updated to MitoCarta2 about a year ago. This is a slightly larger list of about 1100 genes with some reason to be classified as encoding mitochondrial proteins. The addition of about 250 genes to the mitocarta2 list is however unlikely to change the analysis results, so this is not a great problem.

Minor points

- 7) Abstract: the sentence "HRR uncovered a reduced respiratory capacity with glutamate as a fuel substrate accompanied by a substrate shift towards preferred utilization of pyruvate and succinate." Needs to be clarified. I am assuming you mean "... in the malignant prostate tissue".
- 8) Page 5: "Primarily affecting single molecules are passed through a genetic bottleneck..." does not make sense. Please rewrite this sentence.
- 9) Table 1: Should read "Gleason score" not "sore".
- 10) Supplementary Figure 1 caption: The phrase "Two big small peaks" would be better written as "Two short high coverage peaks".
- 11) Page 24 "Cataloge" should be "Catalog".

Reviewer #2 (Remarks to the Author):

In this manuscript Schöpf et al. address a relevant topic, metabolic reprogramming in prostate cancer. Specifically, this work focused in comparing fresh malignant and non-malignant prostate tissues. Those samples were used for high-resolution respirometry studies, for whole mitochondrial DNA sequencing and NGS RNA-sequencing.

One of the major findings of this study was the lower oxidative phosphorylation capacity observed in malignant when compared to benign tissue samples. This lower oxidative capacity was specific of substrates feeding NADH to complex I, concretely malate and glutamate in the presence of ADP (added at saturated concentrations). In addition, those malignant samples were more sensitive to reactive oxygen species (ROS) stress. However, the addition of other substrates, such as pyruvate (complex I) and succinate (complex II) do not show a lower oxidative capacity in malignant samples, even more, the addition of these substrates compensate the lower oxidative capacity observed with malate and glutamate. Therefore, the respiratory capacity of benign and malignant prostate tissue samples was similar when coupled respiration was assessed under the presence of malate, glutamate, pyruvate and succinate and after the titrated addition of FCCP to evaluate electron transfer system capacity under those conditions. These results showed that complex I oxidative capacity is not limited and that the experimental observations must be due to a lower capacity of the glutamate-malate pathway to supply NADH to complex I. These functional observations were associated with a higher presence of mitochondrial DNA (mtDNA) heteroplasmies (HPs) in malignant tissue samples, specifically in the coding regions of the mtDNA. Taking together these results show a clear association among glutamate-malate pathway capacity, ROS susceptibility and mtDNA HPs in malignant tissue samples. In addition, assessment of mRNA expression via NGS showed a differential gene expression profile of nuclear selected genes related to mitochondrial function and metabolism between tissue samples showing a high or a lower glutamate-malate pathway capacity. Therefore, supporting metabolic reprogramming, particularly the pathways related to fatty acid and pyruvate utilization were upregulated and pathways linked to glutamate utilization and metabolism were decreased.

Overall, the results presented in this manuscript are of interest and add new and relevant information to better understand tumor metabolism. However, the current study will improve his impact by clearly demonstrating at the protein level how glutamate metabolism is altered. Are the key enzymes in the glutamate pathway less abundant (protein content)? Or is there any post-translational modification that is decreasing its activity? RNA sequencing data point into a lower protein content but this is not difficult to determine and should be added to the study. Metabolic flux studies will be also of major interest to integrate the observed changes in glutamate metabolism with other intermediary metabolic pathways and therefore better understand tumor metabolism. These studies could be difficult to perform with the experimental setting used in this study and it will require a different experimental approach (cell culture or animal models).

Reviewer #3 (Remarks to the Author):

The manuscript entitled, 'Shift from glutamate to pyruvate and succinate mediated mitochondrial respiration in primary human prostate cancer tissue' claims to have uncovered changes in mitochondrial respiratory substrate preference and mtDNA mutational load in prostate cancer tissue compared with paired non-malignant tissue. Although investigating respiration in paired samples such as these is not novel (see Chekulayev et al. *Biochem Biophys Rep* [2015]), the investigation of prostate tissue is. One important aspect of investigating tumor metabolism in paired samples is a field effect from the malignant tissue on the neighboring histologically normal tissue. This was indeed noted in the paper cited above, where respiration of 'nearby' tissue displayed an intermediate phenotype. In the manuscript under review, the authors do not provide information as to where the paired non-malignant tissue was taken from. It is therefore difficult to be sure that the tissue was far enough away from the malignant lesion to be as 'normal' as possible.

Further major concerns lie in the protocol used for respirometry. The authors use a method that incubates the permeabilized tissue with 500 μ M peroxide for 15 minutes before reduction with catalase. Although an interesting approach, the electron transport chain is highly susceptible to oxidative damage - particularly the Fe-S centres within complex I and III. It is highly likely that measurements of ETC activity after this treatment are compromised, and therefore the comparison of normal and tumor tissue is flawed. A further point is that in order to compare glutamate/malate as substrates before and after peroxide treatment, additional amounts of each should be added.

The authors present the respirometry data in Figure 3 as box and whisker plots with standard deviation. The box and whisker plots appear to be showing median values rather than mean, as suggested in the figure legend. If the whiskers are showing standard deviation, this is not the standard use of this type of plot. This therefore also brings into question the statistical analyses performed.

It would help significantly if the raw data were presented as a table so that the reader can confirm significance when minor changes in values between groups are found to be significantly different. One example of this is Figure 3C.

Finally, the analysis of the mRNA expression data appears erratic, and decreases confidence in the analysis. Figure 8 displays a diagram that highlights the changes of expression that are shown in the heatmap in Figure 7A. The enzymes shown are often incorrect: ACLY is responsible for the irreversible cleavage of citrate to form Acetyl coA and oxaloacetate. Here it is shown in the role of citrate synthase. LDHD is a recently described LDH isoform that has been suggested to convert methylglyoxal to lactate. The widely accepted pyruvate/lactate conversion isozymes are LDHA and B. The IDH enzyme shown is unspecified, but presuming that the authors are showing a TCA cycle, this should be IDH3 (or perhaps IDH2). In Figure 7A, IDH1 is shown, which is cytosolic.

Further minor points:

1. Data not shown is no longer acceptable (page 17, line 1).
2. Figure 2 - labels do not always line up with where the inhibitors/substrates were added - e.g. malonate.
3. Page 19. Is the coverage 10000 fold, rather than 10.085 fold?
4. Page 22. It appears odd to conclude the presence of a gene re-arrangement (ERG) through IHC to determine over-expression of the gene. This approach obviously only provides information on expression, and not the nature of the genetic locus.

Answers to Reviewer Comments

We are very grateful to the reviewers for their valuable and helpful comments and suggestions. Thank you very much for investing your valuable time! We took this as a motivation to continue our analysis and improve our data. We added further supportive data, revised our manuscript carefully and improved the presentation of our data.

New data included:

- 1. Impact of mtDNA mutations on the structure of mt-Complex I. 3D-modeling of CI gene mutations provide a structural bases for the observed shift of respiratory capacities. A visualization of the functionally relevant mutation sites can be viewed online in any JavaScript enabled web browser (<http://www.ruppweb.org/ICM/>).*
- 2. Association of respiratory phenotype tumors with disease-free survival (DSF). In 4 of 5 publicly available prostate cancer cohorts, patient with a tumor exhibiting a respiratory phenotype-like gene expression profile had a shorter DSF.*
- 3. mtDNA copy number analysis. The analysis revealed increased mtDNA/nDNA copy numbers in high-risk prostate cancer.*
- 4. Association of N- to S-pathway respiratory shift with high-risk tumors (Gleason score ≥ 8). Tumors with most severe phenotype are characterized by potentially deleterious, high-level heteroplasmies in CI protein-encoding genes and increased mtDNA copy number.*

With this, we kindly ask for the possibility to resubmit our manuscript for consideration in Nature Communications. Below we provide answers to the questions and comments raised by the reviewers.

Reviewer #1 (Remarks to the Author):

My major concerns are detailed below. My overriding concern is that the authors may not be correctly defining heteroplasmy. Details on that are given in point 1. If I assume that they are correctly defining heteroplasmy, then the reported heteroplasmies are at too small a level to be reasonably causing any functional effect on the mitochondria (point 2).

1) My fundamental concern with this paper is that the observed mtDNA heteroplasmies are all at relatively low levels, 3% to <50% with an average value of only about 10%. Since all of the reported heteroplasmy values (unless I missed something) are below 50% I wonder if the authors are defining heteroplasmy correctly, as the frequency of the allele different from the reference sequence, rCRS. The maximum of 50% raises the possibility that the authors are instead defining the heteroplasmies as minor allele frequencies, which is not the correct way to define it. A heteroplasmic variation from the rCRS of 90% is completely different from a heteroplasmy of 10%. If the heteroplasmy levels are being reported as minor allele frequencies, then this needs to be changed over to the frequency of differences from the rCRS before a correct interpretation of these heteroplasmies can be made.

A: We thank the reviewer for pointing this out! We indeed provided the minor allele frequencies in the previous version, which explains the lower heteroplasmy levels of <50%. We changed the values to the correct heteroplasmy levels by switching to the frequencies of differences to the rCRS, and reanalyzed all subsequent data. After switching to the rCRS-based heteroplasmy levels, the results improved even further, and our statement on the Benign vs Cancer heteroplasmic variants gets underlined even more (see Figure). We updated to the correct HP levels in all tables and figures, respectively.

2) Assuming that the authors are defining the heteroplasmy levels correctly (as differences from the rCRS), then the low heteroplasmy levels they are reporting should not affect mitochondrial function. Experience based on proven pathogenic mtDNA variants has taught us that a rather high threshold of 60-80% heteroplasmy must be passed before mitochondrial function begins to decrease. Indeed, the authors report that the heteroplasmies do not affect OXPHOS function, which is consistent with my expectation from heteroplasmies below the 60-80% heteroplasmy threshold.

A: We misleadingly defined heteroplasmy levels as minor allele frequency in the first version of the manuscript and now carried out a re-analysis based on the frequency of differences from the rCRS. In addition, now included structural analysis revealed an effect of detected mutations on Complex I function (See novel Paragraph "A structural basis for the effect of CI mutations"). Now included in the Discussion: mtDNA heteroplasmy levels up to an allele frequency of 80% may exhibit only a limited functional impact on cellular energy production (Rossignol, Faustin et al. 2003). Although this observation seems well supported by deleterious mtDNA mutations affecting CIV, ATP-synthase or tRNAs, like the A3243G tRNA^{Leu} mutation causing MELAS syndrome or the A8344G tRNA^{Lys} mutation causing MERRF syndrome (Mazat, Letellier et al. 1997, Mazat, Rossignol et al. 2001, Picard et al. 2014), the evidence regarding mutational threshold effects in genes leading to symptomatic CI-associated mitochondrial myopathies is in conflict with this notion. For example, bioenergetics studies on tissues obtained from clinically affected patients suffering from LHON or Leigh's syndrome consistently show "moderate" variant levels of deleterious heteroplasmies (20 - 50%) associated with ~50% decrease in N-pathway capacity (Bai et al. 2000, Kirby et al. 2003, McFarland et al. 2004, Bai and Wong 2005). These findings are in agreement with our study with heteroplasmy levels of 30-60% exhibiting a significant N- to S-pathway shift.

3) The authors report that these heteroplasmies affect glutamate-malate pathways. I see no convincing arguments here for why mutations in the electron transport chain should affect those pathways and not the OXPHOS function that they are directly involved in. The model that they lay out in Figure 8 argues for why altered gene expression can lead to the glutamate-malate pathway differences, but does not really address the point about the effect of apparently low-level heteroplasmies in the mtDNA on this pathway.

A: We thank the reviewer for this remark. Although total OXPHOS capacity in malignant PCa tissue is not affected, the contribution of different substrates for energy generation is significantly changed. Strong support for a causal link between mtDNA mutations and the reduced N-linked respiration detected by HRR comes from the nature and location of the mutations according to the structural data, which are now included: (1) Potentially deleterious mutations in mt-genes encoding CI proteins were accumulated in malignant samples exhibiting a strong N- to S-capacity phenotype. (2) In a high-resolution 3D model of CI, these mutations cluster to the central axis, described as a highly charged channel involved in redox coupling

and the translocation of protons (Baradaran et al. 2013). (3) In-silico 3D structure modeling indicates a significant structural impact by mutations such as those leading to the exchange of a hydrophobic by a polar amino acid.

With gene expression analysis we intend to address the question, whether the observed OXPHOS substrate shift leads to a corresponding adaptation of the expression of upstream metabolizing enzymes, which fuel energy into the respiratory chain either via CI or via CII. The metabolism of glutamate and malate provides electrons to NAD⁺ via various dehydrogenases located in the mitochondria. Reduction of the capacity of this route might trigger a feedback regulation of expression of the involved upstream enzymes. Arguing in the opposite direction, alterations of expression of metabolizing enzymes might facilitate or select for remodeling of OXPHOS pathways.

4) Page 20: No definition was given in the methods section of how "private" heteroplasmies were defined. I presume this was from a comparison against some public database of reported heteroplasmies. This definition can vary greatly depending on the database used. This needs to be included in the methods.

A: We apologize for this lack of clarity in the definitions given by our first manuscript. By using "private" we denominate the occurrence in one tissue only – be it cancer or benign tissue, but not shared. We added in Results: Private mutations, defined as HPs found only in one tissue type, were more frequent in the cancerous samples (Fig. 3A). We added in M&M: Private mutations were defined as HPs found only in one tissue type, as opposed to shared HPs present in both tissue types. All found HPs are listed in SI Table S3, categorized in "private benign", "private cancer" and "shared" HPs.

5) Page 24-25: The transcriptome analysis begins by restricting to 858 genes from the MitoCarta catalog. If the analysis is truly restricted to those 858 MitoCarta genes, then KEGG pathway analysis is not valid here since only a subset of genes involved in mitochondrial functions was tested. The authors need to be clear what null hypothesis was used in the KEGG DAVID analysis. The default null hypothesis for this tool would be all genes, which is not correct here. If only the 858 MitoCarta gene expressions were analyzed, then the null hypothesis should be limited to those 858 genes, not the full gene list. No details are given in the Methods section for this analysis. If the full human gene list was used in the KEGG DAVID analysis, then I see no reason for mentioning the MitoCarta list.

A: We thank the reviewer for this valuable comment. The KEGG DAVID analysis list was restricted to the genes of proteins and enzymes listed in the MitoCarta2 list and used for the entire mRNA expression analysis. KEGG analysis was used solely to group the mt-related genes found to be differentially expressed in our samples into respective pathways without testing for a specific hypothesis. To test for significant differences in gene expression between benign/malignant and mild/severe phenotype samples, a pairwise comparison between groups for each gene listed in the MitoCarta2 catalogue was performed based on the Fragments Per Kilobase Million (FPKM) values (Fig 7D in the new manuscript). To control for type-I error accumulation, p-values were corrected by the Benjamini–Hochberg procedure. This specification is now included in M&M.

6) A related point is that the MitoCarta list was updated to MitoCarta2 about a year ago. This is a slightly larger list of about 1100 genes with some reason to be classified as encoding mitochondrial proteins. The addition of about 250 genes to the MitoCarta2 list is however unlikely to change the analysis results, so this is not a great problem.

A: Transcriptome data were re-analyzed based on the updated list genes encoding mitochondrial proteins MitoCarta2. As expected by the reviewer, this changed analysis results only negligibly.

Minor points

7) Abstract: the sentence "HRR uncovered a reduced respiratory capacity with glutamate as a fuel substrate accompanied by a substrate shift towards preferred utilization of pyruvate and succinate." Needs to be clarified. I am assuming you mean "... in the malignant prostate tissue".

- 8) Page 5: "Primarily affecting single molecules are passed through a genetic bottleneck..." does not make sense. Please rewrite this sentence.
- 9) Table 1: Should read "Gleason score" not "sore".
- 10) Supplementary Figure 1 caption: The phrase "Two big small peaks" would be better written as "Two short high coverage peaks".
- 11) Page 24 "Cataloge" should be "Catalog".

A: all minor points were corrected, changed or rephrased.

Reviewer #2 (Remarks to the Author):

In this manuscript Schöpf et al. address a relevant topic, metabolic reprogramming in prostate cancer. Specifically, this work focused in comparing fresh malignant and non-malignant prostate tissues. Those samples were used for high-resolution respirometry studies, for whole mitochondrial DNA sequencing and NGS RNA-sequencing

One of the major findings of this study was the lower oxidative phosphorylation capacity observed in malignant when compared to benign tissue samples. This lower oxidative capacity was specific of substrates feeding NADH to complex I, concretely malate and glutamate in the presence of ADP (added at saturated concentrations). In addition, those malignant samples were more sensitive to reactive oxygen species (ROS) stress. However, the addition of other substrates, such as pyruvate (complex I) and succinate (complex II) do not show a lower oxidative capacity in malignant samples, even more, the addition of these substrates compensate the lower oxidative capacity observed with malate and glutamate. Therefore, the respiratory capacity of benign and malignant prostate tissue samples was similar when coupled respiration was assessed under the presence of malate, glutamate, pyruvate and succinate and after the titrated addition of FCCP to evaluate electron transfer system capacity under those conditions. These results showed that complex I oxidative capacity is not limited and that the experimental observations must be due to a lower capacity of the glutamate-malate pathway to supply NADH to complex I. These functional observations were associated with a higher presence of mitochondrial DNA (mtDNA) heteroplasmies (HPs) in malignant tissue samples, specifically in the coding regions of the mtDNA. Taking together these results show a clear association among glutamate-malate pathway capacity, ROS susceptibility and mtDNA HPs in malignant tissue samples. In addition, assessment of mRNA expression via NGS showed a differential gene expression profile of nuclear selected genes related to mitochondrial function and metabolism between tissue samples showing a high or a lower glutamate-malate pathway capacity. Therefore, supporting metabolic reprogramming, particularly the pathways related to fatty acid and pyruvate utilization were upregulated and pathways linked to glutamate utilization and metabolism were decreased.

Overall, the results presented in this manuscript are of interest and add new and relevant information to better understand tumor metabolism. However, the current study will improve his impact by clearly demonstrating at the protein level how glutamate metabolism is altered. Are the key enzymes in the glutamate pathway less abundant (protein content)? Or is there any post-translational modification that is decreasing its activity? RNA sequencing data point into a lower protein content but this is not difficult to determine and should be added to the study. Metabolic flux studies will be also of major interest to integrate the observed changes in glutamate metabolism with other intermediary metabolic pathways and therefore better understand tumor metabolism. These studies could be difficult to perform with the experimental setting used in this study and it will require a different experimental approach (cell culture or animal models).

A: We thank the reviewer for these very constructive comments and the succinct and precise summary of our paper. In our study we focused on the association of a respiratory phenotype and the mutations we detected in prostate tissue samples. With now added data on the impact on the structure of CI and the

association of the metabolic phenotype with worse disease-free survival in several independent patient cohorts. This provides strong evidence for a causal link and an impact on malignancy. This is a starting point to dig deeper into the molecular changes and we agree with the reviewer that metabolic flux studies are of primary interest. As the reviewer also noticed, cellular models are appropriate for such studies. In our view, this should be investigated in a separate carry on analyses, beyond our human tissue based study.

Reviewer #3 (Remarks to the Author):

The manuscript entitled, 'Shift from glutamate to pyruvate and succinate mediated mitochondrial respiration in primary human prostate cancer tissue' claims to have uncovered changes in mitochondrial respiratory substrate preference and mtDNA mutational load in prostate cancer tissue compared with paired nonmalignant tissue. Although investigating respiration in paired samples such as these is not novel (see Chekulayev et al. Biochem Biophys Rep [2015]), the investigation of prostate tissue is. One important aspect of investigating tumor metabolism in paired samples is a field effect from the malignant tissue on the neighboring histologically normal tissue. This was indeed noted in the paper cited above, where respiration of 'nearby' tissue displayed an intermediate phenotype. In the manuscript under review, the authors do not provide information as to where the paired non-malignant tissue was taken from. It is therefore difficult to be sure that the tissue was far enough away from the malignant lesion to be as 'normal' as possible.

A: We thank the reviewer for this very important point and we agree that we must be cautious regarding possible field and sampling bias effects. Therefore, benign samples were obtained from regions distant to the tumor site within a region classified as benign by histopathology investigation, in most cases from the contralateral site of the gland. This information is now included in Results. We now also included a control experiment performed with paired benign/benign samples. This experiment revealed (1.) identical HRR results in both paired samples and (2.) no difference in benign samples extracted from different anatomical regions of the gland (SI Fig. S2).

Further major concerns lie in the protocol used for respirometry. The authors use a method that incubates the permeabilized tissue with 500 μ M peroxide for 15 minutes before reduction with catalase. Although an interesting approach, the electron transport chain is highly susceptible to oxidative damage - particularly the Fe-S centres within complex I and III. It is highly likely that measurements of ETC activity after this treatment are compromised, and therefore the comparison of normal and tumor tissue is flawed. A further point is that in order to compare glutamate/malate as substrates before and after peroxide treatment, additional amounts of each should be added.

A: We understand and appreciate the concerns of the reviewer. Although this method has already been used and reported previously (Stadlmann et al. 2002), several control experiments were performed to assess a possible negative impact of the oxidative stress treatment on the function of the ET system (summarized in SI Fig. S1 and described in Results): Analysis of split tissue samples performed with or without the oxidative stress step indicated a consistent difference through all post-treatment substrate/coupling states but no specific targeting of the ET-machinery. O₂-flux differences were measured only for the oxidative stress step, all other SUIT protocol steps remained unchanged. Analysis of respiratory capacities using a protocol without H₂O₂ treatment, now also included (SI Fig. S1), confirmed that the observed alteration in PCa tissue was not compromised by oxidative stress treatment.

The authors present the respirometry data in Figure 3 as box and whisker plots with standard deviation. The box and whisker plots appear to be showing median values rather than mean, as suggested in the figure legend. If the whiskers are showing standard deviation, this is not the standard use of this type of plot. This therefore also brings into question the statistical analyses performed.

A: We thank the reviewer for this comment. All Figures were redesigned and updated. See for example Fig. 2D,E:

It would help significantly if the raw data were presented as a table so that the reader can confirm significance when minor changes in values between groups are found to be significantly different. One example of this is Figure 3C.

A: We agree with the reviewer. In the Figures of the new manuscript, O₂-Fluxes with SDs are shown for all respiratory states.

Finally, the analysis of the mRNA expression data appears erratic, and decreases confidence in the analysis. Figure 8 displays a diagram that highlights the changes of expression that are shown in the heatmap in Figure 7A. The enzymes shown are often incorrect: ACLY is responsible for the irreversible cleavage of citrate to form Acetyl coA and oxaloacetate. Here it is shown in the role of citrate synthase. LDHD is a recently described LDH isoform that has been suggested to convert methylglyoxal to lactate. The widely accepted pyruvate/lactate conversion isozymes are LDHA and B. The IDH enzyme shown is unspecified, but presuming that the authors are showing a TCA cycle, this should be IDH3 (or perhaps IDH2). In Figure 7A, IDH1 is shown, which is cytosolic.

A: We thank the reviewer for pointing out our lack of accuracy. Gene expression was reanalyzed based on the mt-related genes as listed in MitoCarta2. This revealed that important enzymes and transport proteins involved in citrate formation and turnover (e.g. MPC2, PDH, PDK, CS, IDH3) and succinate pathway enzymes were significantly upregulated, which is in line with the findings of metabolite preference in the cancer tissue as shown by the HRR experiments. We updated the figure accordingly (Fig. 7).

Further minor points:

1. Data not shown is no longer acceptable (page 17, line 1).
2. Figure 2 - labels do not always line up with where the inhibitors/substrates were added - e.g. malonate.
3. Page 19. Is the coverage 10000 fold, rather than 10.085 fold?
4. Page 22. It appears odd to conclude the presence of a gene re-arrangement (ERG) through IHC to determine over-expression of the gene. This approach obviously only provides information on expression, and not the nature of the genetic locus.

A: All data, to which we refer are included. Figure 2 was redesigned and improved. Coverage information was improved. Presence or absence of an TMPRSS2-ERG gene rearrangement is often included in histopathological reviews; since this has no relevance for this manuscript these data were omitted.

References:

- Bai, R. K. and L. J. Wong (2005). "Simultaneous detection and quantification of mitochondrial DNA deletion(s), depletion, and over-replication in patients with mitochondrial disease." J. Mol. Diagn **7**(5): 613-622.
- Bai, Y., R. M. Shakeley and G. Attardi (2000). "Tight control of respiration by NADH dehydrogenase ND5 subunit gene expression in mouse mitochondria." Mol Cell Biol **20**(3): 805-815.
- Baradaran, R., J. M. Berrisford, G. S. Minhas and L. A. Sazanov (2013). "Crystal structure of the entire respiratory complex I." Nature **494**(7438): 443-448.
- Kirby, D. M., A. Boneh, C. W. Chow, A. Ohtake, M. T. Ryan, D. Thyagarajan and D. R. Thorburn (2003). "Low mutant load of mitochondrial DNA G13513A mutation can cause Leigh's disease." Ann Neurol **54**(4): 473-478.
- Mazat, J. P., T. Letellier, F. Bedes, M. Malgat, B. Korzeniewski, L. S. Jouaville and R. Morkuniene (1997). "Metabolic control analysis and threshold effect in oxidative phosphorylation: implications for mitochondrial pathologies." Mol Cell Biochem **174**(1-2): 143-148.
- Mazat, J. P., R. Rossignol, M. Malgat, C. Rocher, B. Faustin and T. Letellier (2001). "What do mitochondrial diseases teach us about normal mitochondrial functions...that we already knew: threshold expression of mitochondrial defects." Biochim Biophys Acta **1504**(1): 20-30.
- McFarland, R., D. M. Kirby, K. J. Fowler, A. Ohtake, M. T. Ryan, D. J. Amor, J. M. Fletcher, J. W. Dixon, F. A. Collins, D. M. Turnbull, R. W. Taylor and D. R. Thorburn (2004). "De novo mutations in the mitochondrial ND3 gene as a cause of infantile mitochondrial encephalopathy and complex I deficiency." Ann Neurol **55**(1): 58-64.
- Picard, M., J. Zhang, S. Hancock, O. Derbeneva, R. Golhar, P. Golik, S. O'Hearn, S. Levy, P. Potluri, M. Lvova, A. Davila, C. S. Lin, J. C. Perin, E. F. Rappaport, H. Hakonarson, I. A. Trounce, V. Procaccio and D. C. Wallace (2014). "Progressive increase in mtDNA 3243A>G heteroplasmy causes abrupt transcriptional reprogramming." Proc. Natl. Acad. Sci. U. S. A **111**(38): E4033-E4042.
- Rossignol, R., B. Faustin, C. Rocher, M. Malgat, J. P. Mazat and T. Letellier (2003). "Mitochondrial threshold effects." Biochem. J **370**(Pt 3): 751-762.
- Stadlmann, S., G. Rieger, A. Amberger, A. V. Kuznetsov, R. Margreiter and E. Gnaiger (2002). "H₂O₂-mediated oxidative stress versus cold ischemia-reperfusion: mitochondrial respiratory defects in cultured human endothelial cells." Transplantation **74**(12): 1800-1803.

Reviewers' comments:

Reviewer #3 (Remarks to the Author):

In addressing the highlighted concerns of the reviewers, the manuscript from Schopf and colleagues has significantly improved. However, an increase in mitochondrial mutational rate has already been published (Hopkins et al. *Nature Communications*, 2017), which comprehensively sets out rates and identities of mtDNA mutations in prostate cancer. In the introduction of the current manuscript, the authors propose to extend those findings to identify implications of high rates of mtDNA mutations on mitochondrial function. This they do to some degree using high resolution respirometry and they indeed show alterations in function. However, given that the authors are not the first to show alterations in the mtDNA rate and position information, they must compare their results to the original study, and more importantly why the apparent positions of mutations are different between the studies.

The study continues to have significant limitations, some of which are highlighted by the authors themselves in the discussion. These limitations are significant; focusing on one particular limitation - the respiratory measurements - the use of supramaximal concentrations on isolated mitochondria provide interesting information on the maximal possible respiratory activity of the mitochondria, but not their function within the tumour cell environment, which is important. There are no mechanistic details either that prove that it is these mutations that specifically lead to changes in respiratory function. Compensation for loss of complex I function through increased complex II would require increased non-respiratory NADH-oxidising activity - this should be not only predicted, but shown.

Other major comments

The respiration data are normalized to wet tissue weight. However, mitochondrial weight is altered in prostate cancer (Grupp et al. *Molecular Cancer*, 2013), and particularly in those tumours with a high Gleason score. The higher succinate oxidising activity that the authors noted in this manuscript is also correlated with Gleason score, and therefore suggests that there is little/no change in complex II activity on a per mitochondria basis, while complex I oxidising activity is much more significantly affected than might be predicted from HP rates. Their conclusions, that the tumour tissue are 'transcriptionally programmed to efficiently compensate' loss of complex I activity with an increase in complex II cannot currently be concluded.

The authors suggest that deficiencies in complex IV are likely to affect total respiration more than those in complex I (p10, line 258-onwards). However, their data in Figure 4C shows the those samples with non-synonymous HP in CIII/IV have higher respiration on glutamate/malate than those with mutations in complex I. This either suggests that the rates of HP are tolerated at different levels depending on the position of the complex in the ETC, or that the mutations that are tolerated in CIII and IV affect respiratory activity less. This really should be tested mechanistically - a theme throughout the manuscript that is unfortunately lacking.

Minor comments

Unfortunately, despite the improvements, the manuscript still has errors. Examples include; Page 13, line 331; contrary to the suggestion of the authors, to my knowledge, SLC13A3 has not previously been shown as a mitochondrial transporter of succinate, and is not within the canonical SLC25 mitochondrial transporter family.

Page 13, line 340-onwards; glutamate-driven ATP production is typically defined as glutamine-driven ATP production (i.e. glutaminolysis), and then can include glutaminase (which synthesises glutamate). PRODH has been previously shown to support exogenous proline catabolism, rather than utilizing the proline synthesised from glutamate (Elia et al. *Nature Communications* 2017). Page 13, line 343-onwards; although PDK2, 3 and 4 may inactivate PDH complex, PDK1 is the isozyme that has been extensively characterised as regulating PDH complex flux. Importantly, it is

often observed as being increased in tumours

Reviewer #4 (Remarks to the Author):

This manuscript is a fairly detailed look at mitochondrial function and associated heteroplasmic mutations in prostate cancer. The overwhelming strength of the study is direct measurement of oxidative phosphorylation function as well as mitochondrial mutation in a well characterized clinical cohort with matching pair of benign and malignant prostate tissues. The authors have responded well to all of the reviewer comments. Specific attention is brought to comments regarding the strength of the association and the frequency of heteroplasmic, non-synonymous, mitochondrial mutations. While a minor fraction of heteroplasmic mutation in mitochondria can be a concern in terms of functional effects, the frequencies of heteroplasmy found for non-synonymous mutations in the primary tumor tissues is fairly high and consistent with functional mutations found in other primary tumors. In addition, it is unclear whether or not even a minor fraction of heteroplasmic mutation may have downstream effects on regulatory molecules outside the direct respiratory chain, thereby influencing a cellular phenotype. The demonstration of a correlation of alterations in oxidative phosphorylation with nonsynonymous mutations, coupled by a reasonably high fraction of heteroplasmy in non-synonymous mutations in cancer tissues, argues for a functional effect of these mutations. In addition, the fraction of heteroplasmy found in these primary tissues is consistent with that found in other tumor tissues. Of note there is likely additional contamination from non-tumor supporting cells that would likely result in an underestimate of heteroplasmy in primary tumor tissues in these non microdissected samples, thus the heteroplasmy fractions provided are underestimates of heteroplasmy in primary tumor. Overall, the authors have responded well to these criticisms and this criticism in particular, and this manuscript should provide a valuable addition to the literature demonstrating mitochondrial mutations functional effects in primary human cancers

NATURE COMM-16- 29818A-Z: Response to reviewer comments

We thank the reviewers for investing their time and providing valuable comments and suggestions to improve our manuscript. We responded by performing additional experiments and adding new data that support our hypothesis:

1. HRR experiments in a panel of cell lines with and without rotenone inhibition of CI function show that there is a compensatory N→S pathway shift in the OXPHOS state. These results were added to Fig 4.
2. IHC expression analysis of mt markers for mitochondrial mass, CI and CII, revealed an increase of all markers in tumors and decreased CI/CII ratio in tumors in association with the presence and level of non-synonymous mt-CI gene HPs, which was furthermore in good agreement with increased mtDNA copy numbers in high grade tumors. These results were added in Fig. 6 and SI Figs. S6 and S7.
3. A comparison to previous studies on mtDNA mutations in prostate cancer was performed and added: Table S6 and Fig S8.
4. Description and presentation of metabolic enzyme expression was updated and refined in Fig. 7 and in the text.

Point by point response to reviewer comments and suggestions:

Reviewer 3:

In addressing the highlighted concerns of the reviewers, the manuscript from Schopf and colleagues has significantly improved. However, an increase in mitochondrial mutational rate has already been published (Hopkins et al. Nature Communications, 2017), which comprehensively sets out rates and identities of mtDNA mutations in prostate cancer. In the introduction of the current manuscript, the authors propose to extend those findings to identify implications of high rates of mtDNA mutations on mitochondrial function. This they do to some degree using high resolution respirometry and they indeed show alterations in function. However, given that the authors are not the first to show alterations in the mtDNA rate and position information, they must compare their results to the original study, and more importantly why the apparent positions of mutations are different between the studies.

A: We agree with regards to the importance of the study of Hopkins et al. including almost 400 prostate cancer patients and have now included a more detailed comparison with previous NGS mtDNA mutation studies and discussion of our mtDNA mutation results with regard to published data (Discussion, lines 428-445, SI Table S6 and SI Fig. S8).

The different reported NGS mtDNA mutation studies in prostate cancer differ significantly regarding control samples (blood cells or benign prostate tissue), sample isolation (bulk tissue or microdissected tissue) and variant calling definitions and thresholds (SI Fig. S6). This makes a direct comparison difficult. Our study reports a higher overall HP rate than that of Hopkins et al. We think this is due to different HP thresholds (2% vs $\Delta HF > 0.2$), different control tissue (matched prostate tissue vs blood) and different sampling technique (bulk vs. microdissected tissue). When applying the same threshold ($\Delta HF > 0.2$) to filter for HPs, we see a similar HP rate in our tumor cohort (0.5 vs 0.8, SI Fig. S8). Our rate in benign tissue is 0.1 with this threshold, half of the rate reported by Hopkins et al. We think that this is due to the different control samples used (paired benign tissue vs. blood). It is well known that mtDNA mutations vary across different organs, also within one individual (1, 2) potentially due to tissue-specific positive selection.

The mtDNA variants identified in our study show little overlap to published mtDNA variants in tumors. We found only one mutation in common to a reported prostate cancer mutation in the TCGA PRAD-US cohort. Additionally, a few mutations were reported to be associated with other diseases as listed in RESULTS (lines 198-214). These findings supports the notion of random mtDNA mutagenesis by replication and DNA repair errors as well as a lack of driver mutations as pointed out previously by Ju et al. (2014). This information is included and discussed (lines: 198-214, 435-438)

We see comparison of paired benign and malignant tissue samples and relating mtDNA HPs directly to mitochondrial respiratory function as the specific strength of our study. In these aspects, it is different from all previously published studies. To the best of our knowledge, our study is the first to directly relate the mutational mtDNA landscape of primary human prostate cancer to mitochondrial metabolism and function. Whereas Hopkins et al. focused specifically on mutations located within non-coding areas of the mtDNA e.g. the origins of replication for the heavy strand (OHR), we placed our focus on mutations that we expected to impact mitochondrial function, e.g. mutations affecting the proteins of the mitochondrial machinery. As a main result of our study, we provide compelling evidence that non-synonymous mutations in mt-CI protein genes affect N-pathway OXPHOS capacity and are associated with risk of tumor progression. We added new data supporting this link: IHC analysis of mt-markers shows a significant decrease of the CI/CII ratio in those tumors harboring non-synonymous CI-gene HPs (Fig 4, SI Figs. S6,S7).

The study continues to have significant limitations, some of which are highlighted by the authors themselves in the discussion. These limitations are significant;

We agree and have honestly addressed and discussed limitations of our study. The data we provide here, are by no means the end of the story. We believe strongly however that they are a good hypothesis and starting basis for subsequent and more detailed investigations into the metabolic and gene expression changes described herein and identifying underlying molecular regulations and interactions.

Focusing on one particular limitation - the respiratory measurements - the use of supramaximal concentrations on isolated mitochondria provide interesting information on the maximal possible respiratory activity of the mitochondria, but not their function within the tumour cell environment, which is important.

A: We thank the reviewer for this remark, and we do indeed agree that the use of either isolated mitochondria and/or isolated cells does not address the importance of the tumor cell environment. However, we did not use isolated mitochondria but permeabilized tissue for our experiments. Exactly this method avoids isolation and preselection bias and allows the evaluation of respiratory function of malignant cells within their complex cellular and micro-environmental structure. This technique was developed previously (3, 4) and performed to selectively disrupt the cell membrane while preserving mitochondrial membrane integrity. One limitation of this approach is mentioned in the discussion: the potential dilution of the malignant phenotype in the HRR experiments due to non-malignant cells within the microenvironment.

The use of saturating (not supramaximal) concentrations is needed to test for differences in OXPHOS- and ETS-capacities and to evaluate mitochondrial quality (5). If non-saturating concentrations are used, reactions and enzyme kinetics are limited by substrate and/or oxygen levels creating artificial bottlenecks rather than by real defects or up/downregulation in ETS proteins or other relevant enzymes.

There are no mechanistic details either that prove that it is these mutations that specifically lead to changes in respiratory function.

A: We agree that a final direct proof is desirable. This would need introduction of specific mitochondrial mutations into cellular models to characterize the impact on mitochondrial respiration and metabolism. Whilst this should be feasible, it is not possible within the current timeframe. Thus, we have attempted to address this comment by providing a structural analysis of all mutations suspected to be deleterious using a high-resolution crystal structure of mt Complex I modeling to provide a potential mechanistic rationale for the causal relationship of these mutations to N-linked OXPHOS. Although an experimental wet-lab confirmation for a mechanistic cause is not currently available, the approach of an in-silico analysis based on Complex I crystal structure data was already used by Gopal et al. and was argued to be indicative for a causal relationship due to the nature and location of the mutations within Complex I coding genes (6). To our knowledge, this is the first study in primary human cancer tissue linking deleterious mutations in ETS-subunit coding genes to the OXPHOS profile providing also a structural rationale by in-silico modeling. We also wish to add that an extensive body of literature on pathogenic mtDNA variants relied heavily on the combination of (high-resolution) respirometry with mtDNA sequencing (7-12).

Compensation for loss of complex I function through increased complex II would require increased non-respiratory NADH-oxidising activity - this should be not only predicted but shown.

A: It is not easy to analyze the enzymatic activity of the complex network of NADH producing/consuming enzymes and measurement of NAD/NADH levels alone would not provide a final answer. Further investigation of molecular mechanisms and functional testing of the pathways found to be altered in our study is the logical continuation of our project. However, we honestly think, this is a big project on its own, beyond the scope of the current study and size limitations of this manuscript. Our study is focused on tissue as a disease-near model for primary prostate cancer and further investigations into the molecular mechanisms in our view essentially need to include tissue as a model system as well, for example tissue section cultures, besides cellular models. Availability of such samples is limited. Therefore, these studies need careful planning and a timeframe compatible with a complex logistics for getting fresh patient tissue samples and cannot be done short-term.

The respiration data are normalized to wet tissue weight. However, mitochondrial weight is altered in prostate cancer (Grupp et al. Molecular Cancer, 2013), and particularly in those tumours with a high Gleason score. The higher succinate oxidising activity that the authors noted in this manuscript is also correlated with Gleason score, and therefore suggests that there is little/no change in complex II activity on a per mitochondria basis, while complex I oxidising activity is much more significantly affected than might be predicted from HP rates.

A: We apologize for not being precise and detailed enough on this point. We agree with the reviewer that mt-density is altered in high-grade PCa tissue and that absolute oxygen consumption per mg tissue might be influenced by differences in mt-density. However, we are not primarily focusing on the absolute differences per mg of tissue but instead interested in the qualitative changes in substrate preference and substrate partitioning. We used Flux Control Ratios (FCRs) obtained after normalization for maximal NS-mediated flux to compare the respiratory profiles (Figure 2F). FCRs represent coupling and substrate control independent of mt-content in cells or tissues.

To address increased mitochondrial density as a possible mechanism to achieve higher S-pathway capacity, we performed an IHC study in our prostate sample cohort and assessed immunoreactivity for porin (considered as a mt mass marker), NDURFS4 (marker for CI) and SDHA (marker for CII). The results revealed increase of all three markers in tumors, and, most importantly, an altered CI/CII marker ratio that was significantly decreased in tumors harboring potentially deleterious non-synonymous mutations in mtDNA encoded CI genes. These results indeed point to increased mitochondrial density in tumor cells as a likely mechanism to compensate for N-pathway capacity loss. Although it is a matter of debate, whether a mt-mass is a good indicator of OXPHOS capacity per-se, our IHC results are in agreement with previous reports on increased mt-mass associated with progression (13) and increased immunoreactivity of mt-mass and Complex I-V markers in primary tumors (14). We have included and discussed the new IHC results in the revised manuscript (lines 325-336; 473-480, Fig. 6 E-H, SI Fig. S6-S7).

To uncover other possible mechanisms, we performed a HRR study with a panel of 3 malignant and 2 non-malignant prostate cell lines and applied low concentrations (nM) of rotenone to partially inhibit NADH-linked respiration. In all cell lines, partial inhibition elicited an increased S-pathway OXPHOS capacity. These results indicate mobilization of a S-pathway reserve OXPHOS capacity upon reduction of N-pathway oxidative flux in order to sustain a high OXPHOS capacity. These new data are added and discussed in the revised manuscript (lines 266-273, 483-488; Fig. 4G).

Their conclusions, that the tumour tissue are 'transcriptionally programmed to efficiently compensate' loss of complex I activity with an increase in complex II cannot currently be concluded.

A: We admit that this statement maybe too strong and have rephrased it: "Despite the decrease of N-pathway capacity, the cancer cells seem to be capable to reorganize OXPHOS capacities and metabolism to efficiently compensate this loss by an increase in S-linked respiration" (Lines 489-491).

The authors suggest that deficiencies in complex IV are likely to affect total respiration more than those in complex I (p10, line 258-onwards). However, their data in Figure 4C shows the those samples with non-synonymous HP in CIII/IV have higher respiration on glutamate/malate than those with mutations in complex I. This either suggests that the rates of HP are tolerated at different levels depending on the

position of the complex in the ETC, or that the mutations that are tolerated in CIII and IV affect respiratory activity less. This really should be tested mechanistically – a theme throughout the manuscript that is unfortunately lacking.

A: Our data indeed suggest that HPs in CIII/CIV genes are better tolerated than those in CI genes. As can be seen in Figure 4C, the relative activity with glutamate & malate in samples harboring CIII or CIV mutations is not significantly lower compared to samples harboring no mutations but are significantly decreased when mutations are located in CI genes (lines 448-450). We conclude that (i) already moderate HP levels in genes coding for CI subunits can lead to functional changes, in agreement with other studies (15, 16). We could not perform a more detailed analysis on the effect of mutations in CIII or CIV genes related to HP levels because of the significantly lower numbers compared to mutations in CI genes.

Minor comments

Unfortunately, despite the improvements, the manuscript still has errors. Examples include; Page 13, line 331; contrary to the suggestion of the authors, to my knowledge, SLC13A3 has not previously been shown as a mitochondrial transporter of succinate, and is not within the canonical SLC25 mitochondrial transporter family.

The reviewer is right and we apologize for this sloppiness. The product of SLC13A3 (sodium-dependent dicarboxylate transporter member 3), is a cytoplasmic membrane dicarboxylate carrier. Regardless of its location, it can enhance supply with succinate. In addition, also expression of the mitochondrial membrane dicarboxylate transporter SLC25A10 is elevated in the tumors. This was corrected and updated in RESULTS (line 355) and in Fig. 7.

Page 13, line 340-onwards; glutamate-driven ATP production is typically defined as glutamine-driven ATP production (i.e. glutaminolysis), and then can include glutaminase (which synthesises glutamate).

A: We corrected this sentence.

PRODH has been previously shown to support exogenous proline catabolism, rather than utilizing the proline synthesised from glutamate (Elia et al. Nature Communications 2017).

A: Here we disagree with the reviewer. The results of Elia et al. show that exogenous proline is not required, but an intact proline \leftrightarrow P5C loop supported by PRODH and PYCR1. KD of either one of the two gene products inhibited cell growth and ATP production and affected early metastasis, whereas depletion of proline in the medium did not. We thank the reviewer for pointing to this interesting pathway that enables feeding electrons into CII by oxidizing proline and regenerating it using NADPH. Rechecking our RNAseq expression data revealed that PYCR1 expression is increased in the tumors suggesting enhancement of this path to generate ATP through CII. This information is now included and discussed in the revised manuscript (lines 360-362, 515-518, Fig. 7).

Page 13, line 343-onwards; although PDK2, 3 and 4 may inactivate PDH complex, PDK1 is the isozyme that has been extensively characterised as regulating PDH complex flux. Importantly, it is often observed as being increased in tumours

A: Indeed, we find PDK1 increased in the tumors. This was already contained in Fig.7 and now also listed in the mentioned upregulated genes in RESULTS (line 351)

Literature:

1. Lee HY, Chung U, Park MJ, Yoo JE, Han GR, Shin KJ. Differential distribution of human mitochondrial DNA in somatic tissues and hairs. *Ann Hum Genet.* 2006;70(Pt 1):59-65.
2. Li M, Schroder R, Ni S, Madea B, Stoneking M. Extensive tissue-related and allele-related mtDNA heteroplasmy suggests positive selection for somatic mutations. *Proc Natl Acad Sci U S A.* 2015;112(8):2491-6.
3. Kuznetsov AV, Strobl D, Ruttmann E, Konigsrainer A, Margreiter R, Gnaiger E. Evaluation of mitochondrial respiratory function in small biopsies of liver. *Anal Biochem.* 2002;305(2):186-94.
4. Schopf B, Schafer G, Weber A, Talasz H, Eder IE, Klocker H, et al. Oxidative phosphorylation and mitochondrial function differ between human prostate tissue and cultured cells. *FEBS J.* 2016;283(11):2181-96.
5. Gnaiger E. Capacity of oxidative phosphorylation in human skeletal muscle: new perspectives of mitochondrial physiology. *Int J Biochem Cell Biol.* 2009;41(10):1837-45.
6. Gopal RK, Kubler K, Calvo SE, Polak P, Livitz D, Rosebrock D, et al. Widespread Chromosomal Losses and Mitochondrial DNA Alterations as Genetic Drivers in Hurthle Cell Carcinoma. *Cancer Cell.* 2018;34(2):242-55 e5.
7. Krylova TD, Tsygankova PG, Itkis YS, Sheremet NL, Nevinitsyna TA, Mikhaylova SV, et al. [High resolution respirometry in diagnostic of mitochondrial disorders caused by mitochondrial complex I deficiency]. *Biomed Khim.* 2017;63(4):327-33.
8. Li R, Guan MX. Human mitochondrial leucyl-tRNA synthetase corrects mitochondrial dysfunctions due to the tRNA^{Leu}(UUR) A3243G mutation, associated with mitochondrial encephalomyopathy, lactic acidosis, and stroke-like symptoms and diabetes. *Mol Cell Biol.* 2010;30(9):2147-54.
9. Pecina P, Houstkova H, Mracek T, Pecinova A, Nuskova H, Tesarova M, et al. Noninvasive diagnostics of mitochondrial disorders in isolated lymphocytes with high resolution respirometry. *BBA Clin.* 2014;2:62-71.
10. Vienne JC, Cimetta C, Dubois M, Duburcq T, Favory R, Dessein AF, et al. A fast method for high resolution oxymetry study of skeletal muscle mitochondrial respiratory chain complexes. *Anal Biochem.* 2017;528:57-62.
11. Gasparre G, Kurelac I, Capristo M, Iommarini L, Ghelli A, Ceccarelli C, et al. A mutation threshold distinguishes the antitumorigenic effects of the mitochondrial gene MTND1, an oncojanus function. *Cancer Res.* 2011;71(19):6220-9.
12. Lebon S, Chol M, Benit P, Mugnier C, Chretien D, Giurgea I, et al. Recurrent de novo mitochondrial DNA mutations in respiratory chain deficiency. *J Med Genet.* 2003;40(12):896-9.
13. Grupp K, Jedrzejewska K, Tsourlakis MC, Koop C, Wilczak W, Adam M, et al. High mitochondria content is associated with prostate cancer disease progression. *Mol Cancer.* 2013;12(1):145.
14. Feichtinger RG, Schafer G, Seifarth C, Mayr JA, Kofler B, Klocker H. Reduced Levels of ATP Synthase Subunit ATP5F1A Correlate with Earlier-Onset Prostate Cancer. *Oxid Med Cell Longev.* 2018;2018:1347174.
15. Kirby DM, Boneh A, Chow CW, Ohtake A, Ryan MT, Thyagarajan D, et al. Low mutant load of mitochondrial DNA G13513A mutation can cause Leigh's disease. *Ann Neurol.* 2003;54(4):473-8.
16. McFarland R, Kirby DM, Fowler KJ, Ohtake A, Ryan MT, Amor DJ, et al. De novo mutations in the mitochondrial ND3 gene as a cause of infantile mitochondrial encephalopathy and complex I deficiency. *Ann Neurol.* 2004;55(1):58-64.

REVIEWERS' COMMENTS:

Reviewer #3 (Remarks to the Author):

I would like to thank the authors for responding positively to the points made, and believe that they have addressed all issues remaining.

Response to the Reviewer

REVIEWERS' COMMENTS:

Reviewer #3 (Remarks to the Author):

I would like to thank the authors for responding positively to the points made, and believe that they have addressed all issues remaining.

We thank the reviewer evaluating our revised manuscript again and for his support and encouraging comment.